# Consistent Neural Embeddings through Flow Matching on Attractor-like Neural Manifolds

## Abstract

The primary objective of brain-computer interfaces (BCIs) is to establish a direct connection between neural activity and behavioral actions through neural decoders. Consistent neural representation is crucial for achieving high-performance behavioral decoding over time. Due to the stochastic variability in neural recordings, existing neural representation techniques yield embedding inconsistency, leading to the failure of behavioral decoders in few-trial scenarios. In this work, we propose a novel Flow-Based Dynamical Alignment (FDA) framework that leverages attractor-like ensemble dynamics on stable neural manifolds, which facilitate a new source-free alignment through likelihood maximization. The consistency of latent embeddings obtained through FDA was theoretically verified based on dynamical stability, allowing for rapid adaptation with few trials. Further experiments on multiple motor cortex datasets validate the superior performance of FDA. The FDA method establishes a novel framework for consistent neural latent embeddings with few trials. Our work offers insights into neural dynamical stability, potentially enhancing the chronic reliability of real-world BCIs.

## 1 Introduction

Brain-computer Interfaces (BCIs) establish a direct link between the brain and external devices, presenting great opportunities for improving neural rehabilitation in individuals with paralysis (Willett et al., 2021; Metzger et al., 2023; Willett et al., 2023). However, sustaining long-term decoding performance in chronic implantation is challenging due to non-stationary neural recordings resulting from behavioral variability (Truccolo et al., 2008), physiological changes (Athalye et al., 2017), and device degradation (Woeppel et al., 2021). Addressing this issue requires understanding the neural origin of behavior (Urai et al., 2022; Krakauer et al., 2017). This necessitates methods that can consistently represent neural recordings with latent embeddings to achieve high-performance behavioral decoding over time (Urai et al., 2022; Jazayeri & Ostojic, 2021).

Existing work on neural representation (Schneider et al., 2023; Dabagia et al., 2023; Safaie et al., 2023) have focused on neural latent embeddings, and aligned them for stable long-term neural decoding. Linear methods, such as principal component analysis (PCA) (Degenhart et al., 2020; Yu et al., 2008; Gallego et al., 2018), are used for interpretable latent factors, but often at the cost of performance (Urai et al., 2022). Non-linear methods (Zhou & Wei, 2020; Pandarinath et al., 2018; Prince et al., 2021) based on low-dimensional neural manifolds usually have explicit assumptions on the statistical properties of dynamical latent variables. For instance, NoMAD (Karpowicz et al., 2022) and the source-free alignment (Vermani et al.,

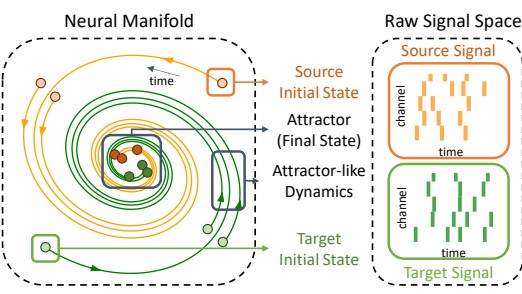

Figure 1: Illustration of attractor-like ensemble dynamics within the neural manifold and raw neural signals with the stochastic variability.

2024) based on seq-VAEs assume Gaussian posteriors for the closed-form calculation of distribution divergences. However, the restrictive assumption does not guarantee consistent neural embeddings, which may limit both their generalizability and interpretability (Schneider et al., 2023). Recently, the pre-trained foundation models based on transformers have been leveraged for shared latent embeddings across abundant sessions and modalities via scaling laws (Ye et al., 2023; Azabou et al., 2023). But these foundation models usually demand extensive data and, more critically, lack interpretability in neural dynamics.

In addition, these existing representation techniques aforementioned may yield inconsistent neural embeddings (Karpowicz et al., 2022; Wang et al., 2023; Vermani et al., 2024) due to stochastic perturbations in neural recordings. Specifically, while they can achieve reasonable performance through alignment with a substantial number of target samples (around 100 trials), their inconsistency can lead to the failure of behavioral decoding over time in few-trial scenarios with no more than 5 target trials. This phenomenon has been empirically validated, as shown in Fig. S4. Hence, the consistency of neural embeddings over time is essential for ensuring controllable deviations under stochastic variability, especially in few-trial scenarios.

Despite the stochastic variability within neural recordings, regions like the motor cortex (Inagaki et al., 2019; Finkelstein et al., 2021; Hira et al., 2013) exhibit a shared low-dimensional manifold when similar tasks are performed. Within this manifold, latent states converge toward similar ones over time, a property known as attractor-like ensemble dynamics (Gonzalez et al., 2019; Khona & Fiete, 2022). This mechanism inspires us to leverage attractor-like ensemble dynamics, where the final similar states serve as neural embeddings. As shown in Fig. 1, this dynamical property enables the rapid adaptation of raw neural signals with stochastic variability, thereby achieving consistent neural embeddings within the neural manifold.

In this work, based on the fact that attractor-like ensemble dynamics is a key property of dynamically stable systems (Bhatia & Szegö, 2002), we propose a novel Flow-Based Dynamical Alignment (FDA) framework to establish such systems with attractor-like dynamics and achieve consistent neural embeddings. Specifically, our FDA approach leverages recent advances in flow matching (Lipman et al., 2023), with the explicit likelihood maximization formulation provided by flows further facilitating a new source-free unsupervised alignment. The consistency of FDA embeddings was theoretically verified through the dynamical stability of neural manifolds, allowing for rapid adaptation with few target trials. Furthermore, extensive experiments on multiple motor cortex datasets validate the superior performance of our FDA over existing approaches. The FDA approach introduces an innovative framework for consistent neural latent embeddings and successfully achieves unsupervised alignment in few-trial scenarios. Our FDA, based on attractor-like ensemble dynamics, offers insights into neural dynamical stability, potentially improving the long-term reliability of real-world BCIs (Dabagia et al., 2023; Fan et al., 2023; Karpowicz et al., 2024). The main contributions of this paper are summarized as follows:

- **Consistent Neural Embeddings**: Flow matching was initially employed on stable neural manifolds using attractor-like ensemble dynamics to achieve consistent neural embeddings. The explicit formulation of likelihood maximization from flow matching provides a novel source-free unsupervised alignment. We establish a new neural representation characterized by consistent embeddings using the mechanism of attractor-like ensemble dynamics.

- **Flow-Based Dynamical Alignment (FDA)**: We propose an innovative framework for Flow-Based Dynamical Alignment (FDA) grounded in consistent neural embeddings. The dynamical stability of FDA is theoretically validated and effectively applied to unsupervised alignment in few-trial scenarios. Our approach has the potential to enhance the chronic reliability of real-world BCIs in the presence of non-stationary neural signals.

- **Experimental Validation**: We extensively validated FDA on several motor cortex datasets (Ma et al., 2023). Results demonstrate that FDA significantly enhances cross-session decoding performance using few target trials. Furthermore, we numerically demonstrate the dynamical stability of neural manifolds based on the Lyapunov exponents.

## 2 RELATED WORK

**Neural Representation for Behavioral Decoding** Previous representation researches have explored various strategies for discovering shared latent neural embeddings over time. Linear dimensionality

reduction methods, such as PCA (Degenhart et al., 2020; Yu et al., 2008; Gallego et al., 2018), were leveraged for interpretable neural state spaces. Non-linear methods (Schneider et al., 2023; Cho et al., 2023) were shown to be effective for representation across trials and sessions. For instance, typical approaches included variational autoencoders (VAEs) with auxiliary variables (Zhou & Wei, 2020; Sani et al., 2021; Klindt et al., 2021) or self-supervised techniques (Liu et al., 2021). More-over, low-dimensional neural dynamics (Pandarinath et al., 2018; Karpowicz et al., 2022), usually assumed to exist within neural manifolds Gallego et al. (2017); Mitchell-Heggs et al. (2023), were utilized as preserved variables under similar behaviors (Safaie et al., 2023). Recently, transformer-based architectures (Liu et al., 2022; Le & Shlizerman, 2022) demonstrated effectiveness for the unified and scalable neural representation. Pre-trained foundation models based on transformer architectures (Azabou et al., 2023; Ye et al., 2023) also achieved desirable latent features across various subjects and sessions. Nonetheless, existing neural representation approaches usually yield dynamical instability due to non-stationary neural signals, resulting in unreliable long-term behav-ioral decoding. Here, we propose a novel framework that leverages attractor-like ensemble dynamics on neural manifolds, ensuring dynamical stability.

**Alignment for Behavioral Decoding** Unsupervised alignment of these neural representa-tions (Dabagia et al., 2023) is crucial for behavioral decoding. Some works focused on directly aligning raw neural signals. For instance, ADAN (Farshchian et al., 2018) and Cycle-GAN (Ma et al., 2023) achieved this through adversarial learning techniques. In addition, latent features such as low-dimensional neural dynamics (Jude et al., 2022; Karpowicz et al., 2022; Wang et al., 2023; Vermani et al., 2024) were aligned across sessions. Few-trial supervised alignment can be accom-plished via fine-tuning with certain pre-trained models (Ye et al., 2023; Azabou et al., 2023). How-ever, existing unsupervised alignment approaches usually lead to unreliable behavioral decoding due to non-stationary neural signals, particularly in few-trial scenarios. In this work, our FDA method achieves source-free unsupervised alignment through likelihood maximization with few target trials, a challenge that most existing unsupervised alignment approaches have not effectively addressed.

## 3 METHODOLOGY

### 3.1 PROBLEM FORMULATION

We define the problem of long-term behavioral decoding based on the unsupervised domain adap-tation (Long et al., 2013). First, we define the domain $\mathcal{D} = \{(x_1, y_1), \ldots, (x_n, y_n)\}$, where $x_i(l)(l = 1, 2, \ldots, m)$ represents the raw neural signal sample from the $l$-th channel in one or more sessions. The signal window has a length of $w$ time points, much smaller than the length of trials, i.e., $x_i(l) \in \mathbb{R}^w$. The first signal window of each trial begins at the initial time point, while the second window starts one step later. $y_i$ denotes the behavioral label corresponding to the $w$-th time step of $x_i$, with $y_i \in \mathbb{R}^d$. The behavioral label is assigned at the $w$-th time step to meet real-time decoding requirements using short-time causal windows and to leverage previous time steps as contextual information effectively.

Based on $\mathcal{D}$, we define the source domain $\mathcal{D}_S$, consisting of signals and labels from one or more sessions: $\mathcal{D}_S = \{(x_1^S, y_1^S), \ldots, (x_{n_S}^S, y_{n_S}^S)\}$. Similarly, the unlabeled target domain $\mathcal{D}_T$ consists of signals from a separate session: $\mathcal{D}_T = \{x_1^T, \ldots, x_{n_T}^T\}$, where $n_T \ll n_S$, and typically only contains signals of few trials. For convenience, we define $\mathbf{x}^S$ and $\mathbf{y}^S$ as the random variables representing neural signals $x_i^S$ and their corresponding labels $y_i^S$ in $\mathcal{D}_S$. Samples $x_j^T$ from $\mathcal{D}_T$ are represented as random variables $\mathbf{x}^T$. We aim to obtain consistent latent embeddings from $\mathcal{D}_S$ and $\mathcal{D}_T$, obtaining high-performance decoders for behavioral labels $\mathbf{y}^T$ associated with $\mathbf{x}^T$.

### 3.2 OVERALL FRAMEWORK

To obtain consistent neural embeddings from non-stationary neural signals, we propose a novel framework that applies flow matching on neural manifolds, constructing a dynamically stable system to achieve attractor-like ensemble dynamics. This framework consists of two phases: pre-training and fine-tuning, as illustrated in Fig. 2.

During the pre-training phase, we establish a continuous normalizing flow on stable neural manifolds using $\mathcal{D}_S$. Specifically, this conditional flow directs noisy latent features toward the target neural manifold using latent dynamics. It is realized through an ordinary differential equation (ODE) with

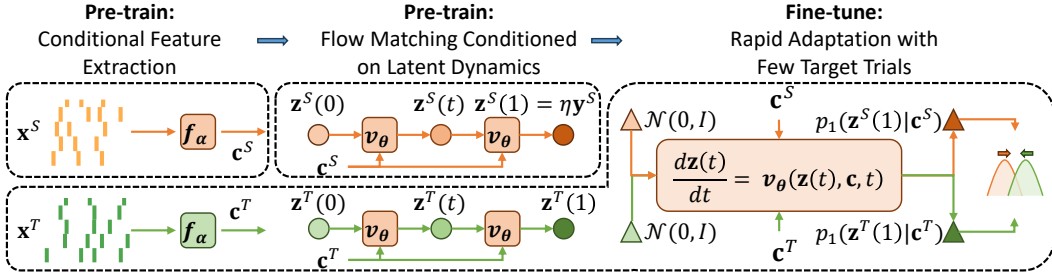

Figure 2: Two phases of the overall FDA framework: pre-training, which involves conditional feature extraction and flow matching conditioned on latent dynamics, and fine-tuning, which enables rapid adaptation with few target trials.

external inputs. The dynamical stability of this flow-based system is ensured by constraining latent state deviations through Lipschitz continuity and regularizing the drift coefficients of latent states, as further explained in Theorem 3.1.

As for fine-tuning, we perform unsupervised rapid adaptation of latent features using few trials from $\mathcal{D}_T$. Compared to some existing flow-based adaptation methods (Gong et al., 2019; Liu et al., 2023a), FDA allows for alignment with fewer target samples. Additionally, based on the explicit computation of log-likelihood using the Fokker-Planck Equation, we propose a novel source-free alignment method through likelihood maximization.

### 3.2.1 FLOW MATCHING ON STABLE AND FLEXIBLE NEURAL MANIFOLDS

During the pre-training phase, we propose a novel framework based on the continuous normalizing flow conditioned on latent dynamics. FDA offers several distinct benefits in obtaining neural latent embeddings. First, flow matching imposes fewer assumptions on the underlying statistics of latent variables, allowing for more flexible modeling on the neural manifolds and improving adaptability to diverse decoding tasks. Second, the flow is governed by a dynamical system with external inputs. Our theoretical analysis demonstrates the dynamical stability of this system, and the empirical results further validate this stability.

**Conditional Feature Extraction Based on Neural Dynamics** We begin by extracting the latent dynamics from $x_i^S$ as conditional features $c_i^S$. For spike signals, a single channel usually records neuron-level activity (Buzsáki, 2004), where the short-term dynamics are relatively limited for similar tasks (Izhikevich et al., 2004). Moreover, inter-channel relationships in spike signals are generally more stable compared to the temporal dynamics, which often exhibit warping (Williams et al., 2020). The above observations are validated, as demonstrated in Fig. 4(c).

Based on these observations and inspired by (Liu et al., 2024), we utilize short-term dynamics to establish conditional feature spaces, and leverage the more stable inter-channel relationships for their coefficients. This approach can flexibly accommodate changes in the number of channels, which is quite common during neuron growth and apoptosis (Degenhart et al., 2020). Specifically, we feed the raw neural signal sequence $x_i^S = [x_i^S(1), \ldots, x_i^S(m)]$, containing tokens from $m$ channels, into a transformer-based network $f_\alpha$ (with parameters $\alpha$) using the classical sinusoidal positional encoding. After processing through multi-head self-attention modules and projection networks, we obtain conditional latent dynamics: $c_i^S = f_\alpha(x_i^S)$, where $c_i^S \in \mathbb{R}^{k_c}$. The detailed architecture is illustrated in Appendix A.1.

**Flow Matching Conditioned on Latent Dynamics** After learning latent dynamics, we establish the continuous normalizing flow conditioned on these dynamics for long-term decoding. Traditional normalizing flows (Chen et al., 2019; Dinh et al., 2022) typically rely on invertible transformations, but these often constrain the representational capacity of networks. Recent researches have utilized continuous normalizing flows (Yang et al., 2019) to alleviate this. For instance, flow matching (Liu et al., 2023a; Lipman et al., 2023; Ma et al., 2024) extends diffusion models, enabling more flexible diffusion paths. Conditional flow matching (Liu et al., 2023b; Zheng et al., 2023; Dao et al., 2023; Isobe et al., 2024; Atanackovic et al., 2024) further incorporates conditional features for modeling conditional distributions. Inspired by these approaches, we adopt conditional flow matching to implement the continuous normalizing flow.

We model the conditional probability $p_t(\mathbf{z}^S(t)|\mathbf{c}^S)$ using a probability flow ODE, where $\mathbf{z}^S(t) \in \mathbb{R}^{k_z}$ denotes the latent states at time point $t \in [0, 1]$, capturing the evolution of $\mathbf{z}^S$ over time. Here, $\mathbf{c}^S$ is the random variable representing conditional features $c_i^S$ ($\mathbf{c}^S = f_\alpha(\mathbf{x}^S)$). Typically, the flows are built on a parameterized flow $\phi_t$ to transform a simple prior distribution $p_0$ (e.g., a multivariate Gaussian) into a more complex one $p_1$: $p_t = [\phi_t]_* p_0$.

To obtain neural embeddings for behavioral decoding, we set $p_0$ as a standard multivariate Gaussian distribution, i.e., $\mathbf{z}^S(0) \sim \mathcal{N}(0, \mathbf{I})$. The target distribution $p_1$, representing the desired neural manifold for behavioral decoding, is defined by the random variable $\mathbf{z}^S(1) = \eta \mathbf{y}^S$, where $\eta \in \mathbb{R}^{k_z \times d}$ is pre-defined with Xavier initialization and remains the same across days. This distribution is denoted as $q(\mathbf{z}^S(1))$, with $\eta^* \in \mathbb{R}^{d \times k_z}$ as the generalized inverse of $\eta$, which serves as weights of the linear decoder $G$ and also satisfies $\eta^* \eta = \mathbf{I}_d$. In the detailed implementation, the flow $\phi_t$ of $p_t(\mathbf{z}^S(t)|\mathbf{c}^S)$ is optimized following conditional flow matching. Within the latent space of $\mathbf{z}^S(t)$, a neural network $v_\theta$ (with parameters $\theta$) is utilized to parameterize the vector field of latent features, allowing for its evolution as follows:

$$\frac{d\mathbf{z}^S(t)}{dt} = v_\theta(\mathbf{z}^S(t), f_\alpha(\mathbf{x}^S), t). \tag{1}$$

Based on Eq. (1), the evolution of $p_t(\mathbf{z}^S(t)|\mathbf{c}^S)$ over time follows the Fokker-Planck Equation:

$$\frac{\partial p_t(\mathbf{z}^S(t)|\mathbf{c}^S)}{\partial t} = -\nabla \cdot \left( p_t(\mathbf{z}^S(t)|\mathbf{c}^S) \, v_\theta(\mathbf{z}^S(t), f_\alpha(\mathbf{x}^S), t) \right). \tag{2}$$

Existing work (Liu et al., 2023b) indicates that the network $v_\theta$ can be optimized using a objective function, which matches the vector field provided by $v_\theta$ with a predefined vector field $u(t)$. To enhance the efficiency of sampling and distribution alignment of latent features, we set the flow path over time as a linear interpolation between the start $\mathbf{z}^S(0)$ and the end $\mathbf{z}^S(1)$:

$$\mathbf{z}^S(t) = (1 - t)\mathbf{z}^S(0) + t\mathbf{z}^S(1). \tag{3}$$

The corresponding vector field of Eq. (3) is $u(t) = \mathbf{z}^S(1) - \mathbf{z}^S(0)$. According to these, the training objective function $\mathcal{L}_{\text{cfm}}(\alpha, \theta)$ can be defined as below:

$$\mathcal{L}_{\text{cfm}}(\alpha, \theta) = \mathbb{E}_{t, p(\mathbf{z}^S(0)), q(\mathbf{z}^S(1))} \left\| v_\theta(\mathbf{z}^S(t), f_\alpha(\mathbf{x}^S), t) - (\mathbf{z}^S(1) - \mathbf{z}^S(0)) \right\|^2, \tag{4}$$

where $\mathbf{z}^S(0) \sim \mathcal{N}(0, \mathbf{I})$, $\mathbf{z}^S(1) = \eta \mathbf{y}^S$. $v_\theta$ only consists of multilayer perceptron (MLP) layers with residual connections, and its detailed architecture is provided in Appendix A.1.

**Dynamical Stability Verification** The dynamical stability (Angeli, 2002) is ensured by two key factors. First, the velocity field in flow matching is constructed using MLPs with Lipschitz-continuous activation functions. These functions ensure that latent state deviations remain stable under external input constraints, as shown in Eq. (7) and Eq. (21). Second, the scale coefficient $\gamma^S$ of latent states is regularized to keep the ratio of latent state deviations between successive time steps below 1. This results in a geometric sequence with a ratio less than 1, causing latent states to gradually converge to similar ones, as presented in Eq. (6) and Eq. (22).

We further analyze the dynamical stability of this system to demonstrate the consistency of latent neural embeddings. Consider any two signal samples $x_i^S$ and $x_j^S$ from $\mathcal{D}_S$, with corresponding conditional features $c_i^S$ and $c_j^S$, and their latent states $\mathbf{z}_i^S(t)$ and $\mathbf{z}_j^S(t)$. We then analyze the upper bound of the distance $\left\| \mathbf{z}_i^S(t) - \mathbf{z}_j^S(t) \right\|$ based on the Euler sampling method. We summarize our theoretical verification in Theorem 3.1 below. Detailed proof can be found in Appendix A.2.

**Theorem 3.1.** *Let the total number of sampling steps in Euler's method be $T$. At the $n$-th step, the time point is $t_n = \frac{n}{T}$. At this point, the distance between any two latent states $z_i^S(t_n)$ and $z_j^S(t_n)$ corresponding to signal samples $x_i^S$ and $x_j^S$ satisfies the following inequality:*

$$\|z_i^S(t_n) - z_j^S(t_n)\| \leq h_z \left( \|z_i^S(0) - z_j^S(0)\|, n \right) + h_c \left( \|c_i^S - c_j^S\| \right), \tag{5}$$

*where $h_z : \mathbb{R}_{\geq 0} \times \mathbb{Z}_{\geq 0} \to \mathbb{R}_{\geq 0}$ is a decreasing function with respect to $n$, given by:*

$$h_z \left( \|z_i^S(0) - z_j^S(0)\|, n \right) = (\mathbf{K}_\gamma)^n \|z_i^S(0) - z_j^S(0)\|, \tag{6}$$

with $0 < \mathbf{K}_\gamma < 1$. Moreover, $h_c : \mathbb{R}_{\geq 0} \to \mathbb{R}_{\geq 0}$ satisfies $h_c\left(\|c_i^S - c_j^S\|\right) \to \infty$ as $\|c_i^S - c_j^S\| \to \infty$. The function $h_c\left(\|c_i^S - c_j^S\|\right)$ can be expressed as:

$$h_c\left(\|c_i^S - c_j^S\|\right) = \left(\sum_{a=1}^{n-1}(\mathbf{K}_\gamma)^a\right)\mathbf{K}_g\|\mathbf{w}_\beta\|\|c_i^S - c_j^S\|, \tag{7}$$

where $\mathbf{K}_g$ is the Lipschitz constant of activation functions in the network $v_\theta$, and $\mathbf{w}_\beta$ represents the weights used for computing shift coefficients (Ma et al., 2024) in $v_\theta$.

Eq. (5) is consistent with the definition of dynamical stability in (Angeli, 2002), demonstrating the dynamical stability of neural latent embeddings.

### 3.2.2 RAPID ADAPTATION WITH FEW TARGET TRIALS

While fine-tuning, existing adaptation methods (Gong et al., 2019; Liu et al., 2023a; Sagawa & Hino, 2022) based on normalizing flows typically consider the source distribution as the starting point, and the target distribution as the endpoint. However, this approach often requires a large number of target samples. Based on flow matching conditioned on latent dynamics, we propose a more efficient strategy with few trials. The pre-trained flow network $v_\theta$ is fixed, while the conditional feature extractor $f_\alpha$ is fine-tuned, aligning the distribution of final decoding embeddings $\mathbf{z}(1)$. Furthermore, the flow path is approximated as a straight line, allowing us to obtain final latent states in just one step. This significantly simplifies the explicit computation of likelihood functions. Unlike ERDiff, which maximizes the log-likelihood upper bound, we propose a direct log-likelihood maximization approach that achieves source-free unsupervised alignment.

**Maximum Mean Discrepancy Alignment with Few Target Trials (FDA-MMD)** When target sizes are small, the alignment based on individual sample probabilities, such as Kullback–Leibler (KL) divergences in GANs, often leads to training instability. In contrast, Maximum Mean Discrepancy (MMD) leverages higher-order moments as overall sample properties, effectively reducing the influence of outliers in limited samples. This is empirically demonstrated in Fig. 4(a). Hence, we adopt a strategy that minimizes MMD distances to align the distributions of latent neural embeddings.

To be specific, taking one-step Euler sampling as an example, the objective function for aligning the final latent state $\mathbf{z}(1)$ based on $\mathcal{D}_T$ is as follows:

$$\min_\alpha \mathcal{L}_{\text{mmd}}(\alpha) = \min_\alpha \left\|\frac{1}{n_S}\sum_{i=1}^{n_S}\varphi(z_i^S(1)) - \frac{1}{n_T}\sum_{j=1}^{n_T}\varphi(z_j^T(1))\right\|_{\mathcal{H}}^2, \tag{8}$$

where $z_i^S(1) = v_\theta(z_i^S(0), 0, f_\alpha(x_i^S))$, and $z_j^T(1) = v_\theta(z_j^T(0), 0, f_\alpha(x_j^T))$. Here, $\mathcal{H}$ represents the reproducing kernel Hilbert space (RKHS), and $\varphi$ is the feature mapping function in that space. In detailed implementation, we utilize a Gaussian kernel to compute the inner product of features.

**Source-Free Alignment via Likelihood Maximization (FDA-MLA)** A notable advantage of flow matching is its explicit modeling of likelihood functions, allowing for accurate computation of distribution transformations. Meanwhile, distribution alignment based on minimizing KL divergences can be seen as maximizing the likelihood in $\mathcal{D}_T$ (Kingma et al., 2019). In cases with few target samples, alignment approaches relying on one-to-one sample mapping tend to fall into sub-optimal solutions (Courty et al., 2017; Kerdoncuff et al., 2021). In contrast, as illustrated in (Wang et al., 2023), likelihood-based alignment is less affected by sample sizes. Moreover, this alignment strategy does not directly depend on source samples, making it suitable for privacy-sensitive data like neural signals, enabling source-free unsupervised alignment.

Specifically, let the signal samples in $\mathcal{D}_T$ be denoted by the random variable $\mathbf{x}^T$, with the corresponding conditional feature $\mathbf{c}^T = f_\alpha(\mathbf{x}^T)$ and the latent embedding $\mathbf{z}^T(1)$ for decoding. In this context, aligning the final latent state of flow between $\mathcal{D}_S$ and $\mathcal{D}_T$ can be achieved by minimizing the KL divergence. This can be accomplished by fine-tuning the parameters $\alpha$ of conditional feature extractor $f_\alpha$:

$$\min_\alpha D_{\text{KL}}\left(p_1(\mathbf{z}^S(1)|f_\alpha(\mathbf{x}^S)) \,\|\, p_1(\mathbf{z}^T(1)|f_\alpha(\mathbf{x}^T))\right) \approx \max_\alpha \log p_1(\mathbf{z}^T(1)|f_\alpha(\mathbf{x}^T)). \tag{9}$$

Since minimizing KL divergences can be approximated as maximizing log-likelihood functions, we can reformulate the above objective function as maximizing the likelihood based on $p_1(\mathbf{z}^T(1)|f_\alpha(\mathbf{x}^T))$, thereby reducing dependence on $\mathcal{D}^S$.

Since the pre-defined flow path is approximated as a straight line, final latent states can be sampled using the one-step Euler method. This also simplifies the computation of likelihood functions for target conditional probabilities. The likelihood of this conditional probability can be explicitly expressed via the change of variables formula (Chen et al., 2018) as:

$$\log p_1(\mathbf{z}^T(1)|f_\alpha(\mathbf{x}^T)) = \log p_0(\mathbf{z}^T(0)|f_\alpha(\mathbf{x}^T)) - \log \left| \det \left( \frac{\partial v_\theta(\mathbf{z}^T(0), 0, f_\alpha(\mathbf{x}^T))}{\partial \mathbf{z}^T(0)} \right) \right|. \quad (10)$$

Considering that $\log p_0(\mathbf{z}^T(0)|f_\alpha(\mathbf{x}^T))$ is independent of $\alpha$, the objective function $\mathcal{L}_{\mathrm{mla}}$ can be further rewritten as below through target neural signals $x_j^T$:

$$\max_\alpha \mathcal{L}_{\mathrm{mla}}(\alpha) \approx \max_\alpha \left( \sum_{j=1}^{n_T} -\log \left| \det \left( \frac{\partial v_\theta(z_j^T(0), 0, f_\alpha(x_j^T))}{\partial z_j^T(0)} \right) \right| \right). \quad (11)$$

More generally, alternative sampling methods can employ the unbiased Hutchinson-trace estimator (Hutchinson, 1989) to estimate the divergence in Eq. (2), facilitating effective alignment through likelihood maximization. Detailed computations are provided in Appendix A.3.

### 3.3 OVERALL LEARNING ALGORITHM

The overall learning algorithm is illustrated in Algorithm 1. During the pre-training phase, we perform supervised optimization of the conditional feature extractor $f_\alpha$ and the flow network $v_\theta$ using $\mathcal{D}_S$, with the objective function $\mathcal{L}_{\mathrm{cfm}}(\alpha, \theta)$. In the fine-tuning phase, the parameter $\theta$ is fixed, and few trials from $\mathcal{D}_T$ are utilized to fine-tune $\alpha$ based on either $\mathcal{L}_{\mathrm{mmd}}(\alpha)$ or $\mathcal{L}_{\mathrm{mla}}(\alpha)$, as described in Section 3.2.2. Further training details are provided in Appendix B.2.

## 4 EXPERIMENTS AND RESULTS

### 4.1 EXPERIMENTAL SETUP

**Datasets** We employed three distinct datasets of extracellular neural recordings from the primary motor cortex (M1) of non-human primates (Ma et al., 2023), as detailed below. Additional information about the datasets can be found in Appendix B.1.
**Center-Out Reaching (CO-C&CO-M)**. Monkeys C and M engaged in a center-out reaching task, where each trial required them to move to one of eight randomized targets, earning a reward for successful reaching.
**Random-Target (RT-M)**. Monkey M performed a random-target task, reaching for three sequentially presented targets at random locations. Each trial started at the workspace center, with a 2.0-second limit to reach each target.
**Data Preprocess and Spilt** We extracted trials from the 'go cue time' to the 'trial end,' followed by digitizing, filtering, and spike detection of the neural signals. The data was then timestamped and smoothed for firing rates in 50 ms bins. Sessions containing approximately 200 trials, along with 2D cursor velocity labels, were used as $\mathcal{D}_S$ for pre-training, while a separate session without labels was used as $\mathcal{D}_T$ for fine-tuning. For few-trial alignment, we used the target ratio $r$ to evaluate the number of target trials from all recorded ones, typically setting $r$ to 0.02, 0.03, 0.04, and 0.06, with 0.02 corresponding to no more than 4 trials. The decoded cursor velocity is assessed using $R^2$ scores, with results averaged over five different random seeds. Additional experimental details and hyper-parameter settings can be found in Appendix B.2.

### 4.2 COMPARATIVE STUDY

**Baselines** The following approaches were utilized as baselines for comparative experiments, with further implementation details provided in Appendix B.3.
**LSTM**(Hochreiter, 1997): Unaligned LSTMs were used as baseline decoders to assess the

Table 1: Comparison of $R^2$ values (in %) of baselines and FDA on CO-M and RT-M datasets($r = 0.02$). The mean and standard deviation over five runs are listed.

| Data | Session | LSTM | CEBRA | ERDiff | NoMAD | Cycle-GAN | **FDA-MLA** | **FDA-MMD** |
|------|---------|------|-------|--------|-------|-----------|-------------|-------------|
| CO-M | Day 0 | $74.18_{\pm4.9}$ | $79.24_{\pm1.38}$ | $82.71_{\pm2.82}$ | $79.77_{\pm4.50}$ | $77.06_{\pm2.21}$ | $\mathbf{84.79}_{\pm0.91}$ | $\mathbf{84.79}_{\pm0.91}$ |
| | Day 8 | $-41.92_{\pm62.49}$ | $-51.92_{\pm12.51}$ | $-65.06_{\pm60.88}$ | $17.15_{\pm6.97}$ | $14.25_{\pm10.29}$ | $23.79_{\pm8.71}$ | $\mathbf{45.23}_{\pm4.44}$ |
| | Day 14 | $-70.57_{\pm16.62}$ | $-1.77_{\pm7.03}$ | $-44.64_{\pm25.37}$ | $12.14_{\pm15.86}$ | $14.20_{\pm11.21}$ | $50.15_{\pm4.85}$ | $\mathbf{55.90}_{\pm3.17}$ |
| | Day 15 | $-51.19_{\pm90.71}$ | $-83.24_{\pm15.03}$ | $-40.72_{\pm19.89}$ | $5.32_{\pm13.11}$ | $9.77_{\pm6.36}$ | $43.59_{\pm3.69}$ | $\mathbf{49.55}_{\pm3.41}$ |
| | Day 22 | $-16.87_{\pm21.57}$ | $-21.10_{\pm7.01}$ | $-81.24_{\pm43.59}$ | $0.16_{\pm6.97}$ | $14.10_{\pm5.22}$ | $\mathbf{33.98}_{\pm7.39}$ | $27.35_{\pm7.34}$ |
| | Day 24 | $-36.71_{\pm26.26}$ | $-10.28_{\pm3.35}$ | $-28.04_{\pm36.96}$ | $14.66_{\pm12.42}$ | $-3.14_{\pm14.96}$ | $48.86_{\pm4.58}$ | $\mathbf{51.28}_{\pm2.53}$ |
| | Day 25 | $-4.15_{\pm29.55}$ | $-64.67_{\pm16.20}$ | $-47.74_{\pm35.31}$ | $-13.74_{\pm29.43}$ | $15.30_{\pm4.99}$ | $31.74_{\pm7.31}$ | $\mathbf{36.79}_{\pm4.12}$ |
| | Day 28 | $0.23_{\pm25.54}$ | $-35.95_{\pm10.54}$ | $-30.18_{\pm40.68}$ | $11.58_{\pm7.58}$ | $0.35_{\pm14.38}$ | $53.27_{\pm7.55}$ | $\mathbf{54.87}_{\pm4.40}$ |
| | Day 29 | $-111.72_{\pm76.49}$ | $-64.32_{\pm15.75}$ | $-64.19_{\pm22.00}$ | $8.96_{\pm16.43}$ | $16.32_{\pm2.99}$ | $36.16_{\pm9.21}$ | $\mathbf{41.26}_{\pm5.70}$ |
| | Day 31 | $-36.40_{\pm20.18}$ | $-81.41_{\pm21.04}$ | $-46.60_{\pm40.86}$ | $-1.96_{\pm49.56}$ | $0.96_{\pm6.68}$ | $56.50_{\pm3.92}$ | $\mathbf{57.10}_{\pm3.24}$ |
| | Day 32 | $-86.33_{\pm86.80}$ | $-40.10_{\pm16.67}$ | $-20.03_{\pm34.99}$ | $9.76_{\pm13.81}$ | $6.18_{\pm13.31}$ | $40.49_{\pm5.69}$ | $\mathbf{44.66}_{\pm4.41}$ |
| RT-M | Day 0 | $77.91_{\pm1.40}$ | $74.86_{\pm1.03}$ | $76.98_{\pm2.62}$ | $74.71_{\pm2.87}$ | $85.19_{\pm2.36}$ | $\mathbf{86.95}_{\pm1.59}$ | $\mathbf{86.95}_{\pm1.59}$ |
| | Day 1 | $63.15_{\pm3.11}$ | $65.97_{\pm2.38}$ | $-9.07_{\pm20.00}$ | $30.92_{\pm19.32}$ | $32.38_{\pm2.33}$ | $71.83_{\pm3.90}$ | $\mathbf{74.32}_{\pm2.25}$ |
| | Day 38 | $-20.62_{\pm32.46}$ | $21.34_{\pm6.71}$ | $1.46_{\pm13.96}$ | $17.61_{\pm10.12}$ | $21.55_{\pm3.36}$ | $55.05_{\pm2.65}$ | $\mathbf{55.39}_{\pm2.80}$ |
| | Day 39 | $-86.31_{\pm47.86}$ | $-36.86_{\pm25.62}$ | $-30.80_{\pm17.92}$ | $12.01_{\pm11.77}$ | $-2.46_{\pm5.32}$ | $38.28_{\pm6.13}$ | $\mathbf{40.44}_{\pm7.31}$ |
| | Day 40 | $-8.36_{\pm17.70}$ | $2.63_{\pm20.16}$ | $-23.79_{\pm25.04}$ | $9.31_{\pm12.01}$ | $22.02_{\pm11.65}$ | $32.16_{\pm8.95}$ | $\mathbf{39.85}_{\pm3.27}$ |
| | Day 52 | $3.12_{\pm11.68}$ | $30.50_{\pm6.94}$ | $-10.33_{\pm7.38}$ | $11.71_{\pm16.90}$ | $10.29_{\pm12.86}$ | $43.35_{\pm4.80}$ | $\mathbf{44.99}_{\pm4.96}$ |
| | Day 53 | $-43.50_{\pm50.26}$ | $42.33_{\pm4.84}$ | $-0.54_{\pm4.55}$ | $10.88_{\pm13.62}$ | $20.70_{\pm1.85}$ | $49.60_{\pm2.53}$ | $\mathbf{50.03}_{\pm4.44}$ |
| | Day 67 | $-148.64_{\pm98.52}$ | $25.09_{\pm13.79}$ | $-11.16_{\pm24.54}$ | $9.53_{\pm20.44}$ | $25.65_{\pm1.59}$ | $42.06_{\pm6.29}$ | $\mathbf{50.29}_{\pm1.07}$ |
| | Day 69 | $-110.99_{\pm93.95}$ | $-38.82_{\pm29.41}$ | $-45.97_{\pm22.79}$ | $-1.49_{\pm7.16}$ | $-5.99_{\pm27.79}$ | $29.52_{\pm7.31}$ | $\mathbf{39.19}_{\pm4.07}$ |
| | Day 77 | $-448.21_{\pm98.67}$ | $-53.79_{\pm21.04}$ | $-2.13_{\pm8.56}$ | $3.81_{\pm16.18}$ | $-1.68_{\pm18.81}$ | $16.19_{\pm9.43}$ | $\mathbf{16.67}_{\pm9.32}$ |
| | Day 79 | $-226.00_{\pm135.06}$ | $-47.01_{\pm13.77}$ | $-0.12_{\pm18.12}$ | $13.12_{\pm22.32}$ | $10.53_{\pm3.33}$ | $\mathbf{39.29}_{\pm6.86}$ | $38.99_{\pm5.70}$ |

challenges of alignment.

**CEBRA**(Schneider et al., 2023): CEBRA served as an advanced tool for discovering generalizable hidden structures and was proved effective across datasets and subjects without alignment.

**ERDiff**(Wang et al., 2023): ERDiff employed diffusion models to reconstruct spatio-temporal structures and aligned them with latent dynamics derived from VAEs.

**NoMAD**(Karpowicz et al., 2022): NoMAD performed alignment within neural manifolds by utilizing LFADS (Pandarinath et al., 2018) to capture the latent dynamics of neural population activities.

**Cycle-GAN**(Ma et al., 2023): Cycle-GAN directly aligned full-dimensional raw signals at each time step through an adversarial approach.

### 4.2.1 EMPIRICAL VALIDATION ON LATENT SPACE STABILITY

To validate the dynamical stability of latent spaces, we measured the maximum Lyapunov exponent (MLE) $\lambda$ of $\mathbf{z}^S(t)$ after pre-training on $\mathcal{D}_S$. The value of $\lambda$ was computed as described in (Wolf et al., 1985), with a non-positive $\lambda$ typically indicating dynamical stability. The detailed definition and computation of $\lambda$ are available in Appendix B.5. Since MLE is based on sequential variables, we compared the $\lambda$ values obtained by FDA with those of ERDiff and NoMAD, focusing on sequential latent factors. The results are presented in Fig. 3(a) and Appendix C.1.1. Consistent with the findings discussed in Theorem 3.1, we found that FDA achieved negative $\lambda$ across all selected datasets, indicating latent space stability. In contrast, both ERDiff and NoMAD frequently exhibited positive $\lambda$, with ERDiff showing greater instability than NoMAD.

### 4.2.2 CROSS-SESSION PERFORMANCE EVALUATION

We further validated the cross-session performance of FDA-MLA and FDA-MMD with limited target trials. First, we conducted experiments with $\mathcal{D}_S$ containing only one session. The average $R^2$ scores, using Day0 as the source session and a target ratio $r$ of 0.02, are presented in Table 1. Full results are available in Appendix C.1.2. FDA-MLA and FDA-MMD consistently outperformed other methods across most sessions. LSTM and CEBRA frequently failed, highlighting the necessity for alignment. Among the alignment baselines, Cycle-GAN and NoMAD performed significantly worse than reported in their original papers due to the scarcity of target samples, as shown in Fig. S4. ERDiff often showed negative scores, aligning with results reported in (Vermani et al., 2024). In contrast, our FDA approach achieved, on average, over 20.00% higher $R^2$ scores on the CO-M and RT-M datasets. While FDA-MLA performed worse than FDA-MMD overall, this difference is understandable given that it is source-free.

We also visualized additional comparisons with the two best baselines, NoMAD and Cycle-GAN. As illustrated in Fig. 3(c), FDA-MLA and FDA-MMD achieved significantly higher average $R^2$ scores

across different values of $r$. These $R^2$ scores were averaged across all target sessions, as well as five random selections of target samples from each session. When $r$ increased to approximately 0.3 (around 60 trials), FDA demonstrated performance comparable to that of Cycle-GAN and NoMAD, as presented in Fig. S4. We observed that FDA-MLA is less affected by $r$, indicating its superiority in few-trial alignment. The overall performance ($r = 0.02$) across all sessions is presented in Fig. 3(d), where FDA-MLA and FDA-MMD demonstrated considerably better $R^2$ on CO-M. Moreover, we conducted comparisons when $\mathcal{D}_S$ included two sessions with $r$ being 0.02. As shown in Fig. 3(b), FDA-MMD outperformed NoMAD and Cycle-GAN on CO-M and RT-M. We also found that FDA can achieve better alignment with more sessions in $\mathcal{D}_S$. Additional visualizations can be found in Appendix C.1.2.

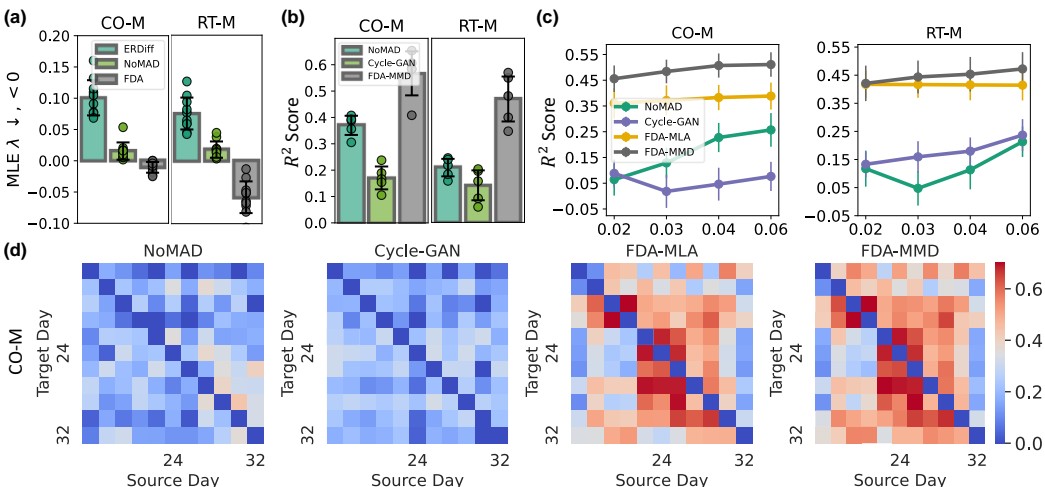

Figure 3: (a) The maximum Lyapunov exponent (MLE) $\lambda$ on CO-M and RT-M datasets. Dots represent the average MLE across five random runs of pre-training for each individual source session. Bar charts denote average MLE across sessions. (b) Comparison of $R^2$ scores for cross-session decoding ($r = 0.02$) with two sessions in $\mathcal{D}_S$. Dots represent the average $R^2$ scores over five runs. (c) Comparison of average $R^2$ scores across target sessions for baselines and FDA under varying $r$ on CO-M and RT-M datasets. (d) Overall average $R^2$ scores ($r = 0.02$) for the same methods as in (c) on the CO-M dataset. Blocks with various colors represent the values of $R^2$.

### 4.3 COMPUTATIONAL EFFICIENCY AND ANALYSIS OF HYPER-PARAMETERS

We compared the computational efficiency of our FDA with that of baselines under similar hardware configurations. The comparison was based on the number of parameters and training time per epoch, which includes pre-training and fine-tuning phases, on CO-C, CO-M, and RT-M. As shown in Table S10, FDA exhibited a higher number of parameters, but it required less training time compared to ERDiff and NoMAD, due to effective training losses and sampling methods. Moreover, the sensitivity analysis of main hyper-parameters in FDA is provided in Appendix C.3.

### 4.4 ABLATION STUDY

#### 4.4.1 ABLATION STUDY ON DIFFERENT ALIGNMENT METHODS

To evaluate the effectiveness of our alignment strategy, we compared FDA with several variants. FDA-t only extracted features using $f_\alpha$ and aligned them through MMD for decoding with a linear decoder. FDA-g used an adversarial approach via Cycle-GAN to align $\mathbf{z}(1)$, while FDA-c applied MMD for aligning $\mathbf{c}$. The average $R^2$ values of CO-M, and RT-M datasets are shown in Table 2. We observed that FDA-MMD consistently outperformed both FDA-t and FDA-g, indicating the advantages of extracting latent features through flows and aligning them via MMD, particularly in scenarios with limited target trials. Additionally, due to flow's accurate modeling of conditional probabilities, FDA-MMD demonstrated more stable performance compared to FDA-c.

Moreover, the $R^2$ curves for FDA-MMD and its variants are shown in Fig. 4(a) and Appendix C.2.1, demonstrating the superior and more stable performance of FDA-MMD. Additionally, as shown in

Fig. 4(b), the negative log-likelihood (NLL) curves and their corresponding $R^2$ values, derived under various $r$ using FDA-MLA, are presented. The results clearly demonstrate that $R^2$ improved as the log-likelihood increased.

Table 2: Comparison of average $R^2$ scores (in %) over sessions on CO-M and RT-M datasets.

| Data | Target Ratio | FDA-t | FDA-g | FDA-c | **FDA-MLA** | **FDA-MMD** |
|---|---|---|---|---|---|---|
| CO-M | 0.02 | $35.94_{\pm 11.36}$ | $35.57_{\pm 6.46}$ | $42.88_{\pm 4.73}$ | $36.05_{\pm 5.84}$ | $\mathbf{45.59_{\pm 5.16}}$ |
| | 0.03 | $41.55_{\pm 8.58}$ | $35.23_{\pm 7.45}$ | $44.69_{\pm 3.72}$ | $37.14_{\pm 5.89}$ | $\mathbf{48.40_{\pm 4.59}}$ |
| | 0.04 | $43.99_{\pm 8.75}$ | $35.25_{\pm 7.66}$ | $46.36_{\pm 4.15}$ | $38.29_{\pm 4.90}$ | $\mathbf{50.71_{\pm 4.68}}$ |
| | 0.06 | $43.78_{\pm 8.09}$ | $34.35_{\pm 8.19}$ | $47.27_{\pm 4.53}$ | $38.85_{\pm 5.23}$ | $\mathbf{51.10_{\pm 4.76}}$ |
| RT-M | 0.02 | $26.61_{\pm 14.04}$ | $40.56_{\pm 7.31}$ | $\mathbf{42.28_{\pm 6.29}}$ | $41.73_{\pm 4.88}$ | $42.08_{\pm 6.31}$ |
| | 0.03 | $27.94_{\pm 12.66}$ | $40.35_{\pm 7.49}$ | $43.77_{\pm 6.05}$ | $41.66_{\pm 4.72}$ | $\mathbf{44.36_{\pm 5.83}}$ |
| | 0.04 | $28.73_{\pm 12.68}$ | $40.04_{\pm 7.62}$ | $44.08_{\pm 6.06}$ | $41.53_{\pm 5.48}$ | $\mathbf{45.35_{\pm 6.15}}$ |
| | 0.06 | $29.21_{\pm 13.72}$ | $39.76_{\pm 7.42}$ | $46.31_{\pm 4.92}$ | $41.44_{\pm 5.42}$ | $\mathbf{47.23_{\pm 5.96}}$ |

### 4.4.2 ABLATION STUDY OF MAIN COMPONENTS

Additional ablation study was conducted, focusing on the main components: the conditional feature extractor $f_\alpha$ and the paths of continuous normalizing flows. FDA-a is the variant incorporating attention mechanisms based on temporal correlations, while FDA-m is the one with $f_\alpha$ implemented by MLPs. For the flow paths, FDA-v and FDA-p are variants using VP and GVP paths (Ma et al., 2024), respectively. The average $R^2$ for each target session achieved by FDA and its variants is shown in Fig. 4(c) and Appendix C.2.2. FDA-MMD and FDA-MLA consistently outperformed FDA-a and FDA-m, highlighting the effectiveness of conditional feature extraction using transformers with inter-channel attention. Additionally, our superior performance over FDA-v and FDA-p further demonstrated the efficiency of the euler sampling method when combined with straight flows.

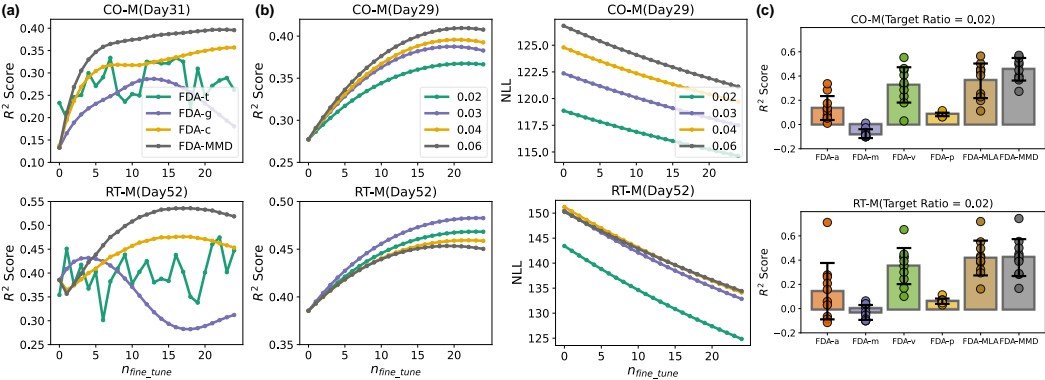

Figure 4: (a) $R^2$ for FDA-t, FDA-g, FDA-c, and FDA-MMD on CO-M (Day31) and RT-M (Day52) with $r = 0.02$. (b) $R^2$ (Left) and the corresponding negative log likelihood (NLL) (Right) on CO-M (Day29) and RT-M (Day52) by FDA-MLA with various target ratios $r$. (c) Comparison of average $R^2$ scores over five runs, achieved by FDA-a, FDA-m, FDA-v, FDA-p, FDA-MLA, and FDA-MMD. Dots represent $R^2$ values for individual session($r = 0.02$). Bar charts denote average $R^2$ across sessions.

## 5 CONCLUSIONS AND LIMITATIONS

In this paper, we establish a new neural representation characterized by consistent neural embeddings based on the mechanism of attractor-like ensemble dynamics. An innovative framework for FDA was proposed on the ground of consistent neural latent embeddings. We achieve the stable dynamics through flow matching on neural manifolds, which enables a novel source-free alignment via likelihood maximization. The dynamical stability of FDA was theoretically verified, allowing for few-trial unsupervised alignment. Extensive experiments on motor cortex datasets demonstrate that FDA significantly enhances decoding performance, offering insights into neural dynamical stability. Our FDA method potentially improves the long-term reliability of real-world BCIs.

This work has several limitations that warrant further investigation. First, the effectiveness of FDA in scenarios such as cross-task or cross-subject alignment needs to be further validated. Second, future studies using clinical data from human subjects could further advance the clinical and chronic applications of BCIs.

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

# A  METHOD

## A.1  DETAILED ARCHITECTURES

We present the detailed architecture of our main modules as follows. The input neural signals have the shape of (Batch size=256, Window size=$w$, Number of channels=$m$). The latent dimensions of conditional features $\mathbf{c}$ are denoted as $k_c$, the dimension of latent states in the continuous normalizing flow as $k_z$. The dropout value is represented as $o_d$. The architectures of $f_\alpha$, and $v_\theta$ can be seen in Table S1.

Table S1: Detailed Architectures of FDA

| $f_\alpha$ | [MSA($k_c$, $n_{head}$), FFN($k_c \times n_{head}$, $k_c$)]×2 |
|---|---|
| $v_\theta$ | MLP($k_z + k_c$, $k_z$, $v_d$)×5 |

Here, we use the term MLP to refer to Multilayer perceptron with residual connections, MSA to represent multi-head self-attention modules, and FFN to indicate feed-forward neural networks.

Moreover, default dimensions $k_c$, $k_z$, the drop-out rate $v_d$, the number of heads $n_{head}$, and the window length $w$ mentioned above are configured as shown in Table S2 according to different datasets.

Table S2: Default Value Setup on Different Datasets

|  | $k_c$ | $k_z$ | $v_d$ | $n_{head}$ | $w$ |
|---|---|---|---|---|---|
| CO-C | 64 | 64 | 0.1 | 8 | 6 |
| CO-M | 32 | 32 | 0.1 | 8 | 5 |
| RT-M | 32 | 32 | 0.1 | 8 | 5 |

## A.2  PROOF OF DYNAMICAL STABILITY IN THEOREM 3.1

- First, consider the iterative relationship between two sampling steps. For example, analyzing the upper bound of $\|z_i^S(t_1) - z_j^S(t_1)\|$ is as follows:

$$\|z_i^S(t_1) - z_j^S(t_1)\| = \|z_i^S(0) + v_\theta(z_i^S(0), 0, f_\alpha(x_i^S)) - z_j^S(0) - v_\theta(z_j^S(0), 0, f_\alpha(x_i^S))\| \quad (12)$$

$$\leq \|z_i^S(0) - z_j^S(0)\| + \|v_\theta(z_i^S(0), 0, f_\alpha(x_i^S)) - v_\theta(z_j^S(0), 0, f_\alpha(x_i^S))\|. \quad (13)$$

In this study, we use an MLP layers with residuals to compose $v_\theta$ as illustrated in (Ma et al., 2024), leading to:

$$v_\theta(z_i^S(0), 0, f_\alpha(x_i^S)) \approx (2 + \gamma_i^S)z_i^S(0) + \beta_i^S, \quad (14)$$

where $\gamma_i^S$ is the scale coefficient, and we assume $0 < \|3 + \gamma_i^S\| < 1$. We only consider the influence of $f_\alpha(x_i^S)$ on $\gamma_i^S$ due to the same sampling time point:

$$\gamma_i^S = g(\mathbf{w}_\gamma f_\alpha(x_i^S) + \mathbf{b}_\gamma). \quad (15)$$

Similarly, $\beta_i^S$ is calculated in the same way:

$$\beta_i^S = g(\mathbf{w}_\beta f_\alpha(x_i^S) + \mathbf{b}_\beta). \quad (16)$$

Thus:

$$v_\theta(z_j^S(0), 0, f_\alpha(x_j^S)) \approx (2 + \gamma_j^S)z_j^S(0) + \beta_j^S. \quad (17)$$

Substituting the expansions of $v_\theta$ into the earlier equation yields:

$$\|z_i^S(t_1) - z_j^S(t_1)\| \leq \|z_i^S(0) - z_j^S(0)\| + \|(2 + \gamma_i^S)z_i^S(0) - (2 + \gamma_j^S)z_j^S(0)\| + \|\beta_i^S - \beta_j^S\| \quad (18)$$

$$\approx \|3 + \gamma_i^S\|\|z_i^S(0) - z_j^S(0)\| + \|\beta_i^S - \beta_j^S\|. \quad (19)$$

Further expanding $\|\beta_i^S - \beta_j^S\|$:

$$\|\beta_i^S - \beta_j^S\| = \|g(\mathbf{w}_\beta f_\alpha(x_i^S) + \mathbf{b}_\beta) - g(\mathbf{w}_\beta f_\alpha(x_j^S) + \mathbf{b}_\beta)\|. \tag{20}$$

Since the activation function $g$ of the MLP is typically a Lipschitz continuous function (e.g., sigmoid function), this simplifies to:

$$\|\beta_i^S - \beta_j^S\| \leq \mathbf{K}_g \|\mathbf{w}_\beta\| \|f_\alpha(x_i^S) - f_\alpha(x_j^S)\| = \mathbf{K}_g \|\mathbf{w}_\beta\| \|c_i^S - c_j^S\|, \tag{21}$$

where $\mathbf{K}_g$ is the Lipschitz constant of the function $g$. Therefore:

$$\|z_i^S(t_1) - z_j^S(t_1)\| \leq \mathbf{K}_\gamma \|z_i^S(0) - z_j^S(0)\| + \mathbf{K}_g \|\mathbf{w}_\beta\| \|c_i^S - c_j^S\|, \tag{22}$$

where $0 < \mathbf{K}_\gamma = \|3 + \gamma_i^S\| < 1$.

- Next, substituting $t_n$ into the above Eq. (22), we obtain the approximate upper bound for $\|z_i^S(t_n) - z_j^S(t_n)\|$:

$$\|z_i^S(t_n) - z_j^S(t_n)\| \leq (\mathbf{K}_\gamma)^n \|z_i^S(0) - z_j^S(0)\| + \left[\sum_{a=1}^{n-1} (\mathbf{K}_\gamma)^a\right] \mathbf{K}_g \|w_\beta\| \|c_i^S - c_j^S\|. \tag{23}$$

Let $h_z(\|z_i^S(0) - z_j^S(0)\|, n) = (\mathbf{K}_\gamma)^n \|z_i^S(0) - z_j^S(0)\|$, where $h_z : \mathbb{R}_{\geq 0} \times \mathbb{Z}_{\geq 0} \to \mathbb{R}_{\geq 0}$ is a decreasing function with respect to $n$.

Let $h_c(\|c_i^S - c_j^S\|) = \left[\sum_{a=1}^{n-1} (\mathbf{K}_\gamma)^a\right] \mathbf{K}_g \|w_\beta\| \|c_i^S - c_j^S\|$, where $h_c : \mathbb{R}_{\geq 0} \to \mathbb{R}_{\geq 0}$, and $h_c(\|c_i^S - c_j^S\|) \to \infty$ as $\|c_i^S - c_j^S\| \to \infty$.

- In summary, the latent space extracted by our method exhibits the dynamical stability defined in (Angeli, 2002).

## A.3  GENERAL COMPUTATION OF LIKELIHOOD IN SECTION 3.2.2

More generally, alternative sampling methods can employ the unbiased Hutchinson-trace estimator (Hutchinson, 1989) to estimate the divergence in Eq. (2). The detailed computation is presented below.

Using the instantaneous change of variables formula (Chen et al., 2018), the log-likelihood $\log p_1(\mathbf{z}^T(1)|f_\alpha(\mathbf{x}^T))$ can be expressed as:

$$\log p_1(\mathbf{z}^T(1)|f_\alpha(\mathbf{x}^T)) = \log p_0(\mathbf{z}^T(0)|f_\alpha(\mathbf{x}^T)) - \int_0^1 \nabla \cdot v_\theta(\mathbf{z}^T(t), f_\alpha(\mathbf{x}^T), t)\, dt, \tag{24}$$

where the latent variable $\mathbf{z}^T(t)$ can be calculated using any sampling method based on Eq. (1). Furthermore, we estimate $\nabla \cdot v_\theta(\mathbf{z}^T(t), f_\alpha(\mathbf{x}^T), t)$ via the unbiased Hutchinson-trace estimator.

Specifically, $\nabla \cdot v_\theta(\mathbf{z}^T(t), f_\alpha(\mathbf{x}^T), t)$ is estimated as:

$$\nabla \cdot v_\theta(\mathbf{z}^T(t), f_\alpha(\mathbf{x}^T), t) = \mathbb{E}_{p(\epsilon)}[\epsilon^\top \nabla v_\theta(\mathbf{z}^T(t), f_\alpha(\mathbf{x}^T), t)\epsilon], \tag{25}$$

where $\nabla v_\theta(\mathbf{z}^T(t), f_\alpha(\mathbf{x}^T), t)$ can be computed via reverse-mode automatic differentiation. The random variable $\epsilon$ satisfies $\mathbb{E}_{p(\epsilon)}[\epsilon] = 0$ and $\mathrm{Cov}_{p(\epsilon)}[\epsilon] = \boldsymbol{I}$.

### A.4 Pseudocode of Flow-Based Dynamical Alignment (FDA) in Section 3.3

---

**Algorithm 1** Flow-Based Dynamical Alignment (FDA)

---

1: **Input:** source domain $\mathcal{D}_S$; target domain $\mathcal{D}_T$; alignment method $align\_m$; pre-defined $\eta$;
2: **Output:** conditional feature extractor $f_\alpha$; continuous normalizing flow network $v_\theta$
3: Initialize $f_\alpha, v_\theta$
4: **Pre-training phase: flow matching conditioned on latent dynamics using $\mathcal{D}_S$:**
5: **for** $iter = 1$ **to** $n_{pre-train}$ **do**
6:     Sample $t$, $\mathbf{z}^S(0) \sim \mathcal{N}(0, \boldsymbol{I})$, $\mathbf{x}^S$, $\mathbf{z}^S(1) = \eta \mathbf{y}^S$;
7:     Update $f_\alpha, v_\theta$ by $\mathcal{L}_{\text{cfm}}(\alpha, \theta)$;
8: **end for**
9: **Fine-tuning phase: few-trial unsupervised alignment based on $\mathcal{D}_S \& \mathcal{D}_T$ or $\mathcal{D}_T$:**
10: **for** $iter = 1$ **to** $n_{fine-tune}$ **do**
11:     **if** $align\_m$ is FDA-MMD: **then**
12:         Sample $\mathbf{x}^S$, $\mathbf{z}^S(0) \sim \mathcal{N}(0, \boldsymbol{I})$ and $\mathbf{x}^T$, $\mathbf{z}^T(0) \sim \mathcal{N}(0, \boldsymbol{I})$; Update $f_\alpha$ by $\mathcal{L}_{\text{mmd}}(\alpha)$;
13:     **else if** $align\_m$ is FDA-MLA: **then**
14:         Sample $\mathbf{x}^T$, $\mathbf{z}^T(0) \sim \mathcal{N}(0, \boldsymbol{I})$; Update $f_\alpha$ by $\mathcal{L}_{\text{mla}}(\alpha)$;
15:     **end if**
16: **end for**
17: **return** $f_\alpha, v_\theta$.

---

## B Experimental Details

### B.1 Dataset Description

**CO-C&CO-M**(Ma et al., 2023). Monkeys C and M conducted a center-out (CO) reaching task while holding an upright handle. Monkey C utilized its right hand, whereas Monkey M used its left. Each trial commenced with the monkey positioning its hand at the center of the workspace. After a random delay, one of eight evenly spaced outer targets arranged in a circle was displayed. The monkey then maintained its position through a variable pause until hearing an auditory go cue. To earn a liquid reward, the monkey needed to reach the outer target within 1.0 second and sustain its hold for 0.5 seconds.

**RT-M**(Ma et al., 2023). Monkey M also participated in a random-target (RT) task, where it reached for sequences of three targets shown in random locations on the screen. This task utilized the same apparatus as the CO reaching task. Each trial started with the monkey placing its hand at the center of the workspace, followed by the sequential presentation of three targets. The monkey had 2.0 seconds to move the cursor to each target after seeing it. Due to the random positioning of the targets, the cursor trajectory varied with each trial.

**Preprocess Process**. For all datasets, we extracted trials from the 'go cue time' to the 'trial end.' Next, we processed the neural signals by digitizing, applying a bandpass filter (250-5000 Hz), and detecting spikes using thresholds based on root-mean square activity. The data was then timestamped and smoothed with a Gaussian kernel to compute firing rates over 50 ms bins.

### B.2 Training Details

The main configurations for model training included the learning rate, weight decay parameters of the Adam optimizer, batch sizes, number of iterative epochs during pre-training and fine-tuning phases. Details of these hyperparameters are provided in Table S3 and Table S4, respectively.

### B.3 Baseline Implementation

**CEBRA**(Schneider et al., 2023). CEBRA is a sophisticated machine-learning approach aimed at analyzing and compressing time series data, particularly in the context of behavioral and neural studies. It excels at revealing hidden structures in data variability and has been effectively applied to decode neural activity in the mouse brain's visual cortex, allowing for the reconstruction of what the subject has seen. The code can be accessed at https://github.com/AdaptiveMotorControlLab/cebra.

Table S3: Detailed Pre-training Setup

|  | Learning Rate | Weight Decay | Epochs | Batch Size |
|---|---|---|---|---|
| CO-C | 2e-3 | 1e-5 | 3500 | 256 |
| CO-M | 2e-3 | 1e-5 | 3500 | 256 |
| RT-M | 2e-3 | 1e-5 | 3500 | 256 |

Table S4: Detailed Fine-tuning Setup

|  | Learning Rate | Weight Decay | Epochs | Batch Size |
|---|---|---|---|---|
| CO-C | 1e-4 | 1e-5 | 25 | 256 |
| CO-M | 1e-4 | 1e-5 | 25 | 256 |
| RT-M | 1e-4 | 1e-5 | 25 | 256 |

**ERDiff**(Wang et al., 2023). ERDiff introduces a method that utilizes diffusion models to extract latent dynamic structures from the source domain and subsequently recover them in the target domain using maximum likelihood alignment. Empirical evaluations on both synthetic and neural recording datasets indicate that this approach surpasses others in effectively preserving latent dynamic structures over time and across individuals. The code can be accessed at https://github.com/yulewang97/ERDiff.

**NoMAD**(Karpowicz et al., 2022). NoMAD utilizes the latent manifold structure present in neural population activity to create a reliable connection between brain activity and motor behavior. It shows the capability to achieve accurate and highly stable behavioral decoding over long durations, thus eliminating the necessity for supervised recalibration. In this study, we implemented NoMAD using the LFADS code found at https://github.com/arsedler9/lfads-torch/tree/main, which may lead to some differences from the original implementation.

**Cycle-GAN**(Ma et al., 2023). Cycle-GAN aligned the distributions of full-dimensional neural recordings, stabilizing the original decoding model without the need for recalibration. Evaluations of Cycle-GAN alongside a related approach (ADAN) on multiple monkey and task datasets reveal that Cycle-GAN outperforms in maintaining BCI accuracy robustly over time without additional training. Since this study employs the same datasets, we directly implement the publicly available code from https://github.com/limblab/adversarial_BCI.

### B.4 Validation Details

Specifically, during the validation after fine-tuning phases, we employed neural signals $\mathbf{x}^T$ from the target domain, which were not leveraged during the fine-tuning phase, to evaluate the efficacy of our alignment approach.

This evaluation is based on the decoding performance based on $R^2$ scores. We first sample $\mathbf{z}^T(1)$ using the one-step Euler based on $\mathbf{z}^T(0)$: $\mathbf{z}^T(1) = v_\theta(\mathbf{z}^S(0), 0, f_\alpha(\mathbf{x}^S))$. The predicted target label $\tilde{\mathbf{y}}^T$ are computed as below: $\tilde{\mathbf{y}}^T = \eta^* \mathbf{z}^T(1)$. $R^2$ scores are further obtained between $\tilde{\mathbf{y}}^T$ and actual $\mathbf{y}^T$.

### B.5 Lyapunov Thoery

The stability described above can be quantified using the Lyapunov function (Angeli, 2002), which can also be estimated through the maximum Lyapunov exponent (MLE). The maximum Lyapunov exponent $\lambda$ can be defined based on the latent state $\mathbf{z}(t)$ as follows: $\lambda = \lim_{t \to \infty} \lim_{|\delta\mathbf{z}(0)| \to 0} \frac{1}{t} \ln \frac{|\delta\mathbf{z}(t)|}{|\delta\mathbf{z}(0)|}$.

A non-positive MLE often indicates the stability of dynamical systems, achieving stable dynamical latent features (Wolf et al., 1985). Here, we estimated the MLE $\lambda$ of $z_i$ based on the method in (Wolf et al., 1985) to evaluate the stability of dynamical latent features extracted from $\mathcal{D}_S$ after the pre-training phase. The detailed calculation of $\lambda$ is available below.

The stability defined in (Angeli, 2002) can be determined using a Lyapunov function $V(z)$: given an equilibrium point $z^*$ of the system,
$V(z^*) = 0$,
$\dot{V}(z^*) = 0$,
$V(z) > 0$ for all $z \neq z^*$,
$\dot{V}(z) < 0$ for all $z \neq z^*$.

It is known that $V(z) = \frac{1}{2} z^T z$ is one of the functions that meet the conditions. However, directly calculating complex $V(z)$ can be difficult. Therefore, we used the method based on (Wolf et al., 1985) to estimate the stability of $z(t)$ as follows:

**Step 1:**
Select $N$ sample points, denoted one as $z_1(t_0)$, find $j$ such that $j = \arg\min_k \|z_1(t_0) - z_k(t_0)\|$, and let $L_0(t_0) = \|z_1(t_0) - z_j(t_0)\|$.

**Step 2:**
Find $t_i$, for a given constant $\epsilon$, such that $t_0 \leq t < t_i$, $L_0(t) \leq \epsilon$; $L_0(t_i) > \epsilon$. Let $L_0' = L_0(t_i)$. Continue with $z_1(t_i)$ as the next sample point following Step 1.

**Step 3:**
The maximum Lyapunov exponent(MLE) $\lambda$ is approximately as follows:

$$\lambda \approx \frac{1}{N\Delta t} \sum_{s=1}^{M} \log_2\left(\frac{L_0'}{L_0(t_0)}\right),$$

where $\Delta t$ is the time step interval and $M$ is the number of steps in a single orbit.

## C ADDITIONAL RESULTS

### C.1 COMPARATIVE STUDY

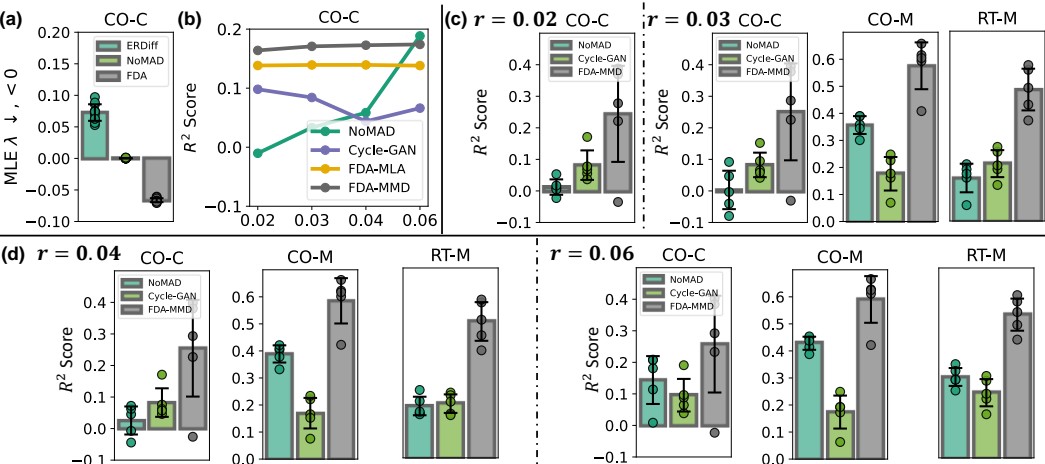

Figure S1: (a) The maximum Lyapunov exponent (MLE) $\lambda$ achieved by ERDiff, NoMAD, and FDA is displayed for the CO-C dataset. Dots in different colors represent the average MLE from individual sessions. (b) Average $R^2$ scores for NoMAD, Cycle-GAN, FDA-MLA, and FDA-MMD are presented under varying values of $r$ on CO-C. (c) and (d): $R^2$ scores for cross-session decoding ($r = 0.02, 0.03$ (c) and $r = 0.04, 0.06$ (d)) when $\mathcal{D}_S$ contains two sessions, obtained from NoMAD, Cycle-GAN, and FDA-MMD, are shown. Dots in different colors represent the average $R^2$ scores for different $\mathcal{D}_S$.

### C.1.1 LATENT SPACE STABILITY

To validate the dynamical stability of latent spaces, we measured the maximum Lyapunov exponent (MLE) $\lambda$ of $\mathbf{z}^S(t)$ after pre-training on $\mathcal{D}_S$. The value of $\lambda$ was computed as described in (Wolf et al., 1985), and the results of CO-C is shown in Fig. S1(a).

We also visualized all maximum Lyapunov exponents (MLE) achieved by ERDiff, NoMAD, and FDA across target sessions. As shown in Fig. S2(a), FDA consistently achieved negative MLEs in most cases, aligning with the average MLE results. This underscores the dynamical stability of its pre-trained latent spaces, in contrast to ERDiff and NoMAD.

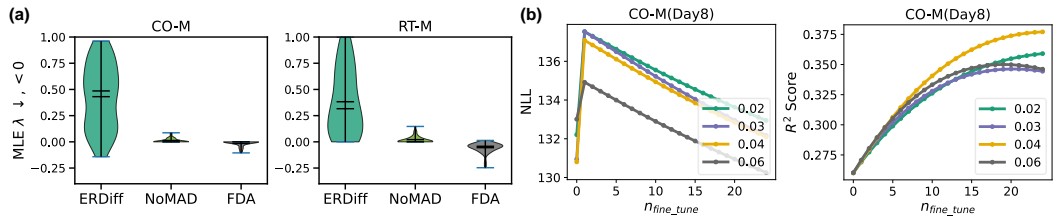

Figure S2: (a) Violin plot of all maximum Lyapunov exponents (MLE) $\lambda$ achieved by ERDiff, NoMAD, and FDA on the CO-M and RT-M datasets. (b) Negative log likelihood (NLL) (Left) and the corresponding $R^2$ (Right) curves on CO-M (Day8) by FDA-MLA with various target ratios $r$.

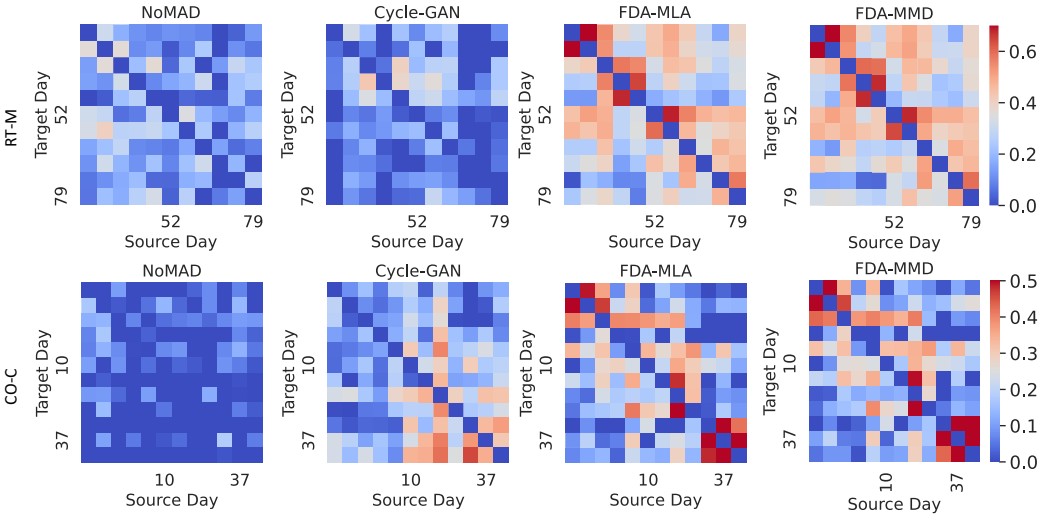

Figure S3: Overall performance of average $R^2$ scores ($r = 0.02$) for NoMAD, Cycle-GAN, FDA-MLA, and FDA-MMD are demonstrated on RT-M, and CO-C datasets. Blocks with various colors represent the corresponding values of $R^2$.

### C.1.2 CROSS-SESSION PERFORMANCE

We verified the cross-session performance of FDA with limited target trials. First, we conducted experiments with $\mathcal{D}_S$ containing only one session. The full average $R^2$ scores on the CO-C dataset, using Day0 as the source session and a target ratio $r$ of 0.02, are presented in Table S5.

In addition, as illustrated in Fig. S1(b), FDA achieved significantly higher average $R^2$ scores across different values of $r$. The overall performance of average $R^2$ on RT-M, and CO-C datasets is presented in Fig. S3.

More comparisons on all datasets when $\mathcal{D}_S$ included two sessions when $r$ equals 0.02, 0.03, 0.04, and 0.06 are shown in Fig. S1(c) and (d).

Table S5: Comparison of $R^2$ values (in %) of baselines and FDA on the CO-C dataset($r = 0.02$). The mean and standard deviation over five runs are listed.

| Data | Session | LSTM | CEBRA | ERDiff | NoMAD | Cycle-GAN | FDA-MLA | FDA-MMD |
|---|---|---|---|---|---|---|---|---|
| CO-C | Day 0 | $86.65_{\pm 1.18}$ | $87.86_{\pm 0.98}$ | $\mathbf{88.69}_{\pm 0.74}$ | $87.99_{\pm 3.45}$ | $84.54_{\pm 1.32}$ | $81.63_{\pm 2.88}$ | $81.63_{\pm 2.88}$ |
| | Day 1 | $16.50_{\pm 33.14}$ | $18.87_{\pm 6.82}$ | $-5.63_{\pm 8.67}$ | $-2.71_{\pm 23.32}$ | $8.05_{\pm 9.86}$ | $49.13_{\pm 5.03}$ | $\mathbf{50.84}_{\pm 5.32}$ |
| | Day 2 | $-9.08_{\pm 42.85}$ | $\mathbf{44.73}_{\pm 14.03}$ | $-8.65_{\pm 16.02}$ | $9.80_{\pm 9.55}$ | $15.35_{\pm 11.34}$ | $36.25_{\pm 5.60}$ | $34.28_{\pm 5.35}$ |
| | Day 3 | $-101.23_{\pm 137.77}$ | $\mathbf{24.47}_{\pm 7.60}$ | $-0.60_{\pm 0.51}$ | $-3.89_{\pm 26.45}$ | $5.40_{\pm 7.21}$ | $7.54_{\pm 4.52}$ | $8.49_{\pm 3.85}$ |
| | Day 9 | $-19.67_{\pm 42.14}$ | $7.79_{\pm 23.55}$ | $-4.10_{\pm 14.87}$ | $-1.46_{\pm 33.06}$ | $18.37_{\pm 7.71}$ | $\mathbf{38.02}_{\pm 7.84}$ | $33.22_{\pm 7.69}$ |
| | Day 10 | $-69.13_{\pm 81.17}$ | $14.64_{\pm 3.55}$ | $0.73_{\pm 13.38}$ | $-3.87_{\pm 11.33}$ | $\mathbf{20.30}_{\pm 8.84}$ | $1.21_{\pm 2.61}$ | $0.76_{\pm 1.26}$ |
| | Day 14 | $-75.51_{\pm 48.00}$ | $-12.97_{\pm 41.24}$ | $-13.82_{\pm 22.93}$ | $-0.19_{\pm 20.93}$ | $2.67_{\pm 14.34}$ | $\mathbf{22.99}_{\pm 7.08}$ | $16.40_{\pm 8.49}$ |
| | Day 15 | $-76.54_{\pm 53.78}$ | $-12.95_{\pm 27.23}$ | $-18.32_{\pm 36.06}$ | $2.87_{\pm 12.97}$ | $\mathbf{19.55}_{\pm 16.31}$ | $9.80_{\pm 15.59}$ | $15.35_{\pm 12.25}$ |
| | Day 16 | $-184.19_{\pm 90.10}$ | $-9.18_{\pm 30.96}$ | $-6.83_{\pm 11.62}$ | $7.56_{\pm 11.54}$ | $6.70_{\pm 11.45}$ | $5.09_{\pm 8.98}$ | $\mathbf{11.04}_{\pm 6.03}$ |
| | Day 36 | $-81.78_{\pm 69.21}$ | $-30.76_{\pm 30.03}$ | $-1.08_{\pm 0.70}$ | $-6.09_{\pm 27.94}$ | $-9.40_{\pm 16.54}$ | $-4.81_{\pm 6.74}$ | $\mathbf{1.00}_{\pm 2.62}$ |
| | Day 37 | $-112.64_{\pm 73.02}$ | $-21.54_{\pm 29.56}$ | $-2.60_{\pm 6.19}$ | $6.58_{\pm 14.01}$ | $8.76_{\pm 6.63}$ | $3.08_{\pm 9.33}$ | $\mathbf{15.95}_{\pm 5.73}$ |
| | Day 38 | $-35.98_{\pm 45.69}$ | $-7.36_{\pm 16.60}$ | $-6.53_{\pm 9.63}$ | $-19.89_{\pm 41.54}$ | $12.17_{\pm 7.03}$ | $-2.77_{\pm 8.46}$ | $\mathbf{12.95}_{\pm 0.92}$ |

To explore the differences in results between Monkey C and Monkey M, we analyzed the cross-session performance of FDA-MMD with greater target ratios $r$. As shown in Table S6, although FDA-MMD initially performed worse on CO-C, its performance improved significantly and became comparable to RT-M when $r$ exceeded 0.3 (approximately 60 trials). Additionally, we observed larger deviations per session on CO-C. This suggests that the difference arises from instability caused by outliers, which notably impacted performance when $r$ was small.

Table S6: Comparison of average $R^2$ values (%) across sessions for FDA-MMD on the CO-C, CO-M, and RT-M datasets ($r = 0.02$). The average standard deviations over five runs per session are also reported.

| $r$ | 0.02 | 0.03 | 0.04 | 0.06 | 0.1 | 0.2 | 0.3 | 0.4 | 0.5 | 0.6 |
|---|---|---|---|---|---|---|---|---|---|---|
| CO-C | $16.40_{\pm 5.40}$ | $17.08_{\pm 7.53}$ | $17.27_{\pm 8.58}$ | $17.41_{\pm 7.66}$ | $28.18_{\pm 5.36}$ | $42.61_{\pm 5.23}$ | $50.12_{\pm 6.90}$ | $54.87_{\pm 5.05}$ | $55.05_{\pm 5.71}$ | $56.00_{\pm 4.88}$ |
| CO-M | $45.59_{\pm 5.15}$ | $48.40_{\pm 4.59}$ | $50.71_{\pm 4.68}$ | $51.10_{\pm 4.76}$ | $57.90_{\pm 2.68}$ | $62.20_{\pm 2.41}$ | $65.16_{\pm 2.53}$ | $66.38_{\pm 2.44}$ | $66.78_{\pm 2.48}$ | $67.32_{\pm 3.32}$ |
| RT-M | $42.08_{\pm 6.31}$ | $44.36_{\pm 5.83}$ | $45.35_{\pm 6.15}$ | $47.23_{\pm 5.96}$ | $52.15_{\pm 4.16}$ | $53.66_{\pm 3.35}$ | $55.28_{\pm 2.89}$ | $56.45_{\pm 2.89}$ | $56.53_{\pm 2.55}$ | $57.93_{\pm 2.39}$ |

Additionally, we observed that the worst $R^2$ score occurred on different days for each method. This variability may stem from the different criteria used for optimal alignment. For instance, FDA-MLA exhibited an abnormal increase in NLL during the initial fine-tuning epochs on Day 8 (CO-M), as shown in Fig. S2(b). In contrast, other methods, such as NoMAD based on KL divergences and LSTM without alignment, did not show this phenomenon on the same day, leading to the worst performance of FDA-MLA while others did not experience such an issue.

### C.1.3 Cross-session Performance under Different Latent Dimensions

To determine the appropriate latent dimensions, we conducted experiments on NoMAD and CEBRA under varying latent dimensions. As shown in Table S7 and Table S8, we selected the latent dimensions for NoMAD and CEBRA as 16 and 32, respectively, based on their better performance. For ERDiff, we set the latent dimension to 8, following the default settings mentioned in the original paper due to its application to similar datasets.

Table S7: Average $R^2$ scores across target sessions of NoMAD on CO-M and RT-M datasets under different latent dimensions.

| Latent Dimension | 12 | 16 | 32 | 48 |
|---|---|---|---|---|
| CO-M | $4.97_{\pm 8.29}$ | $\mathbf{6.40}_{\pm 6.22}$ | $3.69_{\pm 7.00}$ | $-6.21_{\pm 8.70}$ |
| RT-M | $3.42_{\pm 8.78}$ | $\mathbf{11.74}_{\pm 6.42}$ | $8.27_{\pm 10.02}$ | $2.42_{\pm 9.21}$ |

Table S8: Average $R^2$ scores across sessions of CEBRA on CO-M and RT-M datasets under different latent dimensions.

| Latent Dimension | 16 | 32 | 48 |
|---|---|---|---|
| CO-M | $-1.34_{\pm 11.69}$ | $\mathbf{1.14}_{\pm 14.47}$ | $0.85_{\pm 12.61}$ |
| RT-M | $-53.01_{\pm 14.49}$ | $\mathbf{-45.48}_{\pm 12.51}$ | $-49.21_{\pm 14.71}$ |

### C.1.4 ZERO-SHOT CROSS-SESSION PERFORMANCE

Additionally, we compared the zero-shot cross-session performance of NoMAD without alignment, Cycle-GAN without alignment, and FDA without alignment, with detailed results presented in Table S9. FDA without fine-tuning outperformed the baselines, which we attribute to the dynamical stability of its pre-trained latent spaces. Furthermore, performance in few-trial scenarios continued to improve after fine-tuning. In summary, the combination of dynamical stability and fine-tuning contributes to FDA's better performance in few-trial scenarios.

Table S9: Comparison of $R^2$ values (in %) across target sessions (where the $R^2$ scores for each session are averaged over five random runs with different sample selections) of baselines and FDA without alignment on CO-M and RT-M datasets.

| Data | NoMAD w/o alignment | Cycle-GAN w/o alignment | **FDA w/o alignment** | FDA-MLA | FDA-MMD |
|---|---|---|---|---|---|
| CO-M | $-121.47_{\pm 77.80}$ | $-126.84_{\pm 23.82}$ | $16.23_{\pm 9.43}$ | $36.05_{\pm 5.84}$ | $45.59_{\pm 5.15}$ |
| RT-M | $-74.06_{\pm 49.94}$ | $-3.42_{\pm 5.55}$ | $38.15_{\pm 8.21}$ | $41.73_{\pm 4.88}$ | $42.08_{\pm 6.31}$ |

### C.1.5 PERFORMANCE WITH DIFFERENT TARGET RATIOS $r$

To further evaluate the performance of FDA under different target ratios $r$, we gradually increased $r$ from 0.02 to 0.6. The $R^2$ scores for NoMAD, Cycle-GAN, and FDA are shown in Fig. S4. In particular, Cycle-GAN and NoMAD exhibited significantly lower performance (approximately five times worse) with fewer target samples. However, as $r$ increased to around 0.3 (approximately 60 trials), their performance became comparable to that of FDA-MLA and FDA-MMD.

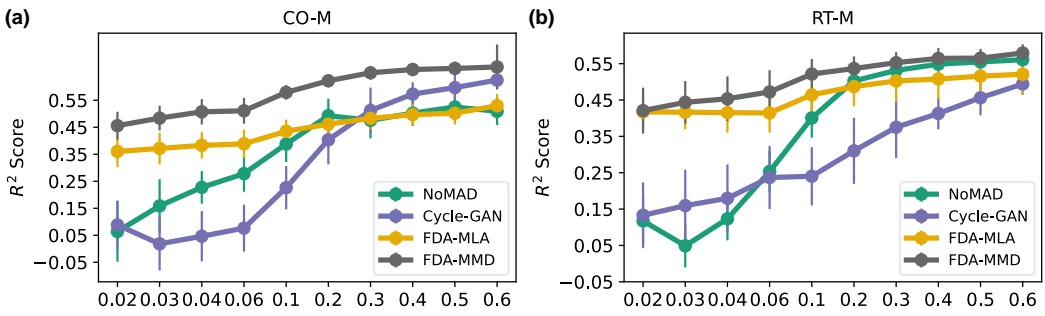

Figure S4: Comparison of $R^2$ scores across target sessions (where the $R^2$ scores for each session are averaged over five random runs with different sample selections) for NoMAD, Cycle-GAN, FDA-MLA, and FDA-MMD under different target ratios $r$ on the (a) CO-M and (b) RT-M datasets.

Additionally, we examined the $R^2$ curves across target sessions for FDA-MMD and Cycle-GAN on the CO-M dataset. As shown in Fig. S5, both methods exhibited fluctuating $R^2$ curves at small target ratios. However, as the target ratio increased, the fluctuations were alleviated. With the exception of

a few sessions, $R^2$ scores generally decreased across most target sessions. We attribute this trend to the reduced influence of certain outliers in scenarios with few target samples.

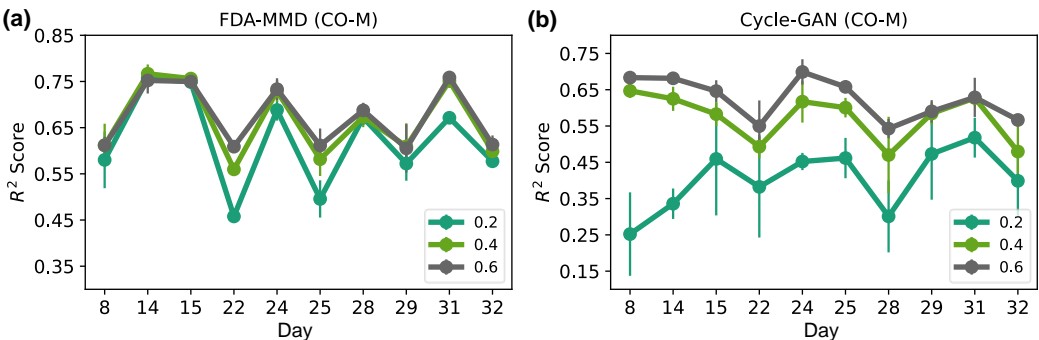

Figure S5: $R^2$ curves across target sessions for (a) FDA-MMD and (b) Cycle-GAN under different target ratios $r$ (0.2, 0.4, and 0.6) on the CO-M dataset.

### C.1.6   COMPUTATIONAL EFFICIENCY

We compared the computational efficiency of our methods with that of ERDiff, Cycle-GAN, and NoMAD. The comparison was based on the number of parameters and training time per epoch, which includes pre-training and fine-tuning, on CO-C, CO-M, and RT-M. As shown in Table S10, FDA-MLA and FDA-MMD exhibited a higher number of parameters. However, they required less training time compared to ERDiff and NoMAD, which can be attributed to effective training losses and sampling methods.

Table S10: Computational Efficiency of Baselines and FDA

| | Method | ERDiff(Wang et al., 2023) | Cycle-GAN(Ma et al., 2023) | NoMAD(Karpowicz et al., 2022) | FDA-MLA | FDA-MMD |
|---|---|---|---|---|---|---|
| | Parameter Number (M) | 0.04 | 0.03 | 0.05 | 0.07 | 0.07 |
| Time(s) | CO-C | 0.39 | 0.05 | 1.05 | 0.14 | 0.14 |
| | CO-M | 1.14 | 0.02 | 1.03 | 0.13 | 0.14 |
| | RT-M | 0.49 | 0.02 | 1.04 | 0.10 | 0.10 |

## C.2 ABLATION STUDY

### C.2.1 DIFFERENT ALIGNMENT METHODS

To evaluate the effectiveness of our alignment strategy, we compared FDA with several variants. FDA-t only extracted features using $f_\alpha$ and aligned them through MMD for decoding with a linear decoder. FDA-g used an adversarial approach via Cycle-GAN to align $\mathbf{z}(1)$, while FDA-c applied MMD for aligning $\mathbf{c}$. The average $R^2$ values of CO-C dataset are shown in Table S11.

Moreover, the $R^2$ curves for FDA-MMD and its variants are shown in Fig. S6(a). Additionally, as shown in Fig. S6(b), the negative log-likelihood (NLL) curves and their corresponding $R^2$ values, derived under various $r$ using FDA-MLA, are presented.

Table S11: Average cross-session $R^2$ scores (%) for CO-C

| Data | Target Ratio | FDA-t | FDA-g | FDA-c | **FDA-MLA** | **FDA-MMD** |
|------|--------------|-------|-------|-------|-------------|-------------|
| CO-C | 0.02 | $-0.33_{\pm 0.29}$ | $13.19_{\pm 9.06}$ | $18.25_{\pm 7.30}$ | $16.39_{\pm 6.30}$ | $13.84_{\pm 5.41}$ |
| | 0.03 | $-0.30_{\pm 0.34}$ | $13.07_{\pm 9.06}$ | $18.49_{\pm 7.38}$ | $17.08_{\pm 6.53}$ | $13.93_{\pm 4.79}$ |
| | 0.04 | $-0.32_{\pm 0.28}$ | $13.06_{\pm 8.89}$ | $18.64_{\pm 7.43}$ | $17.27_{\pm 6.58}$ | $13.94_{\pm 5.64}$ |
| | 0.06 | $-0.23_{\pm 0.25}$ | $13.19_{\pm 8.94}$ | $18.60_{\pm 7.10}$ | $17.41_{\pm 6.66}$ | $13.82_{\pm 5.45}$ |

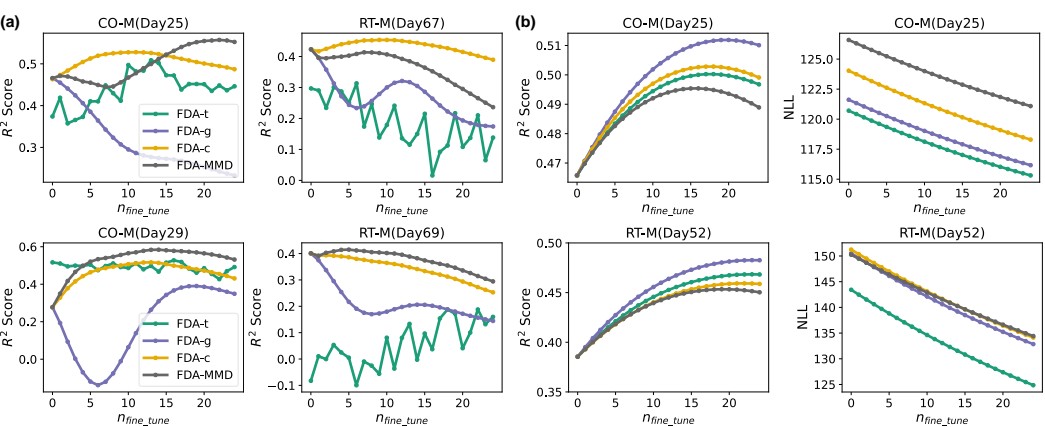

Figure S6: (a) $R^2$ curves for FDA-t, FDA-g, FDA-c, and FDA-MMD are shown on CO-M (Day25, Day29) and RT-M (Day67, Day69) with $r$ being 0.02. (b) Curves for $R^2$ (Left) and the corresponding negative log likelihood (NLL) (Right) on CO-M (Day25) and RT-M (Day52), obtained by FDA-MLA, are visualized under distinct target ratios $r$.

### C.2.2 MAIN COMPONENTS

The average $R^2$ for each target session achieved by FDA and its variants based on main components is shown in Fig. S7.

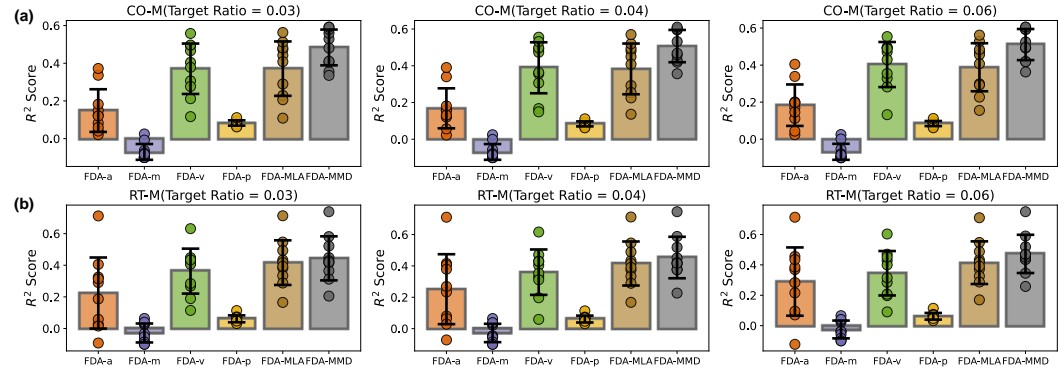

Figure S7: Average $R^2$ scores across each target session, achieved by FDA-a, FDA-m, FDA-v, FDA-p, FDA-MLA, and FDA-MMD, are displayed on CO-M (a) and RT-M (b) datasets with $r$ being 0.03, 0.04, and 0.06. Dots with different colors represent $R^2$ values for individual sessions.

## C.3 HYPER-PARAMETER SENSITIVITY ANALYSIS

The main hyper-parameters of our FDA method include the signal window size ($w$), the dimensions of conditional features and latent states($k_{c,z}$), and the number of euler sampling steps $n_{euler}$ when the target ratio $r$ equals 0.02. For convenience, we set $k_c$ and $k_z$ to be the same. The results of their sensitivity analysis using FDA-MMD on CO-M, and RT-M datasets are shown in Table S12, Table S13, and Table S14.

Table S12: Average $R^2$ scores for different datasets with varying $w$.

| $k_c$ | 4 | 5/6 | 7 | 8 |
|---|---|---|---|---|
| CO-M | $43.91_{\pm 4.68}$ | $45.59_{\pm 5.15}$ | $48.38_{\pm 4.98}$ | $\mathbf{49.07}_{\pm 5.11}$ |
| RT-M | $40.77_{\pm 5.46}$ | $42.08_{\pm 6.31}$ | $40.54_{\pm 7.74}$ | $\mathbf{46.73}_{\pm 3.83}$ |

Table S13: Average $R^2$ scores for different datasets with varying $k_c$.

| $k_c$ | 24 | 32 | 48 | 72 |
|---|---|---|---|---|
| CO-M | $\mathbf{48.00}_{\pm 5.68}$ | $45.59_{\pm 5.15}$ | $45.63_{\pm 4.77}$ | $45.03_{\pm 4.84}$ |
| RT-M | $\mathbf{44.02}_{\pm 5.01}$ | $42.08_{\pm 6.31}$ | $39.48_{\pm 5.51}$ | $43.91_{\pm 4.34}$ |

Table S14: Average $R^2$ scores for different datasets with varying $n_{euler}$.

| $n_{euler}$ | 1 | 2 | 4 | 10 |
|---|---|---|---|---|
| CO-M | $\mathbf{45.59}_{\pm 5.15}$ | $45.32_{\pm 5.14}$ | $43.19_{\pm 5.34}$ | $41.71_{\pm 5.37}$ |
| RT-M | $42.08_{\pm 6.31}$ | $\mathbf{42.14}_{\pm 6.17}$ | $40.33_{\pm 6.12}$ | $38.99_{\pm 6.23}$ |

