# OpenReview forum: "Consistent Neural Embeddings through Flow Matching on Attractor-like Neural Manifolds"
_ICLR.cc/2025/Conference — Submitted to ICLR 2025_

### Official Review · Reviewer_EhkK · 2024-10-21

**Soundness:** 2
**Presentation:** 2
**Contribution:** 3
**Rating:** 6
**Confidence:** 3

**Summary:**

The authors proposed a flow-based framework to align neural embeddings across days. They benchmarked their model against five existing models, three of which focus on BCI alignment: ERDiff (based on diffusion), NoMAD (based on LFADS), and Cycle-GAN (based on Cycle-GAN). Their proposed model consistently outperformed the others across various benchmarks.

**Strengths:**

1. The alignment of raw neural signals or latent embeddings is crucial and necessary for real-world BCI applications.

2. To the best of my knowledge, this is the first study to apply flow-matching to BCI signal alignment.

3. Extensive benchmarking was conducted against state-of-the-art models.

4. The writing is clear and easy to follow.

**Weaknesses:**

1. I am concerned about the evaluation of NoMAD and Cycle-GAN. In a highly relevant submission (https://openreview.net/forum?id=LNp7KW33Cg), which almost certainly comes from the same group, the R² of NoMAD and Cycle-GAN is nearly six times better than what is reported in this paper.

2. This paper uses the exact same dataset as the Cycle-GAN paper (https://elifesciences.org/articles/84296). In the Cycle-GAN paper, the average R² is above 50% (Figure 3A), consistent with the companion submission I mentioned earlier, but in this paper, the evaluation shows an average R² below 10% (Table 1). Why is there such a significant discrepancy?

3. It is inappropriate to cherry-pick results in the presentation. Main Table 1 and Supplementary Table 8 are nearly identical. However, the authors included nine sessions per monkey in Table 1 but omitted two sessions that had the worst performances (Day 22 and 25 in CO-M; Day 40 and 77 in RT-M). Highlighting the best two or three sessions is one thing, but removing the worst two or three sessions is something entirely different.

**Questions:**

1. I am still confused about the term "attractor-like." The authors mentioned it in the context of "*utilizing attractor-like ensemble dynamics (Gonzalez et al., 2019), a representation mechanism for encoding stimuli in the brain.*" I reviewed the referenced paper, but found that "attractor" is only mentioned once: "*Second, attractor-like mechanisms ensure the persistence of representations over short periods of time (days), even if the animals are not exposed to the task or if the circuit is perturbed by lesions.*" I believe additional explanation of what you mean by "attractor-like" in this context would be helpful.

2. Azabou et al., 2023a and Azabou et al., 2023b are the same paper.

---

> ### Author Response · Authors · 2024-11-22
> **Summary of Response to Reviewer EhkK**
>
> Thank you for the thorough read of our manuscript and insightful suggestions.
>
> The performance difference between NoMAD and Cycle-GAN in our work and the referenced paper (https://openreview.net/forum?id=LNp7KW33Cg) is primarily due to the different numbers of target samples used for alignment. The alignment of the referenced paper depends on a substantially larger number of samples (approximately 100 trials). In contrast, our paper aims to enhance alignment for few trials (no more than 5) using a flow-based framework. Therefore, the poor performance of NoMAD and Cycle-GAN in few-trial scenarios highlights the critical need for improvements in alignment, which our flow-based framework effectively addresses. Our flow-based alignment contributes to facilitating rapid adaptation under stochastic variability with few trials in realistic BCI scenarios.
>
> In the following responses, we will address these concerns point-by-point. Thank you for pointing out this unclear point, and we have modified the manuscript to make it clearer.

---

> ### Author Response · Authors · 2024-11-22
> **Response to Weaknesses(1&2) of Reviewer EhkK**
>
> ### Weaknesses:
>
> 1. **I am concerned about the evaluation of NoMAD and Cycle-GAN. In a highly relevant submission(https://openreview.net/forum?id=LNp7KW33Cg), which almost certainly comes from the same group, the R² of NoMAD and Cycle-GAN is nearly six times better than what is reported in this paper.**
>
>    Thank you for this questions. The performance gap of NoMAD and Cycle-GAN was due to **the different numbers of target samples used for alignment**. Specifically, our study only used a small number of target samples (no more than 5 trials) to demonstrate the superiority of our approach, while the paper you mentioned utilized a much larger number of target samples (around 100 trials), therefore demonstrating much better performance.
>
>    To further validate this, we conducted a more detailed analysis on $R^2$ scores achieved by NoMAD and Cycle-GAN under varying target ratios. The results, averaged across target sessions on the CO-M and RT-M datasets, are presented in the two tables below. We observed that the performance of NoMAD and Cycle-GAN degrated as $r$ decreased. The two tables below have been added to Figure S4 in Appendix C.1.5, and this point has been elaborated in Section 4.2.2 (Lines 424-427 on Page 8) with the following new sentence:
>
>    _"Among the alignment baselines, Cycle-GAN and NoMAD performed significantly worse than reported in their original papers due to the scarcity of target samples, as shown in Figure S4."_
>
>
>    **$R^2$ Scores of Cycle-GAN Across Different Target Ratios $r$ on CO-M and RT-M Datasets**
>
>    | $r$ | 0.02 | 0.03 | 0.04 | 0.06 | 0.1 | 0.2 | 0.3 | 0.4 | 0.5 | 0.6 | 0.75 |
>    |:---:|:-----:|:-----:|:-----:|:-----:|:---:|:---:|:---:|:---:|:---:|:---:|:---:|
>    | CO-M   | 8.83 ± 5.37 | 1.83 ± 9.86 | 4.64 ± 9.32 | 7.65 ± 8.68 | 22.62 ± 8.02 | 40.34 ± 9.08 | 51.21 ± 8.46 | 57.27 ± 4.38 | 59.68 ± 4.80 | 62.46 ± 2.97 | 62.44 ± 6.04 |
>    | RT-M   | 13.30 ± 4.54 | 15.95 ± 9.86 | 17.97 ± 9.31 | 23.68 ± 8.95 | 24.02 ± 8.02 | 31.03 ± 9.08 | 37.51 ± 8.46 | 41.30 ± 4.38 | 45.62 ± 4.80 | 49.41 ± 2.97 | 55.43 ± 5.90 |
>
>    **$R^2$ Scores of NoMAD Across Different Target Ratios $r$ on CO-M and RT-M Datasets**
>
>    | $r$ | 0.02 | 0.03 | 0.04 | 0.06 | 0.1 | 0.2 | 0.3 | 0.4 | 0.5 | 0.6 |
>    |:---:|:-----:|:-----:|:-----:|:-----:|:---:|:---:|:---:|:---:|:---:|:---:|
>    | CO-M | 6.40 ± 6.21 | 15.88 ± 9.89 | 22.75 ± 6.03 | 27.80 ± 6.68 | 38.79 ± 6.70 | 49.28 ± 6.29 | 47.56 ± 6.68 | 50.23 ± 4.81 | 52.49 ± 5.10 | 50.90 ± 5.05 |
>    | RT-M | 11.74 ± 6.42 | 4.90 ± 5.97 | 12.32 ± 5.93 | 25.34 ± 5.70 | 40.05 ± 5.41 | 50.26 ± 2.79 | 53.15 ± 2.99 | 54.87 ± 2.54 | 55.48 ± 2.23 | 56.10 ± 3.34 |
>
>    In addition, we added extra descriptions to better clarify the few-trial scenarios based on a small number of target samples in the
>    Introduction(Lines 61-66 on Page 2), which reads:
>
>      _"In addition, these existing representation techniques aforementioned may yield inconsistent neural embeddings due to stochastic perturbations in neural recordings. Specifically, while they can achieve reasonable performance through alignment with a substantial number of target samples (around 100 trials), their inconsistency can lead to the failure of behavioral decoding over time in few-trial scenarios with no more than 5 target trials. This phenomenon has been empirically validated, as shown in Figure S4."_
>
> 2. **This paper uses the exact same dataset as the Cycle-GAN paper (https://elifesciences.org/articles/84296). In the Cycle-GAN paper, the average R² is above 50% (Figure 3A), consistent with the companion submission I mentioned earlier, but in this paper, the evaluation shows an average R² below 10% (Table 1). Why is there such a significant discrepancy?**
>
>    Thanks for raising this question. As mentioneded in the response to the previous question, this performance difference was due to **the different numbers of target samples used for alignment**. Specifically, our study only used a small number of target samples (no more than 5 trials), while the paper you mentioned (https://elifesciences.org/articles/84296) utilized a much larger number of target samples (around 100 trials).
>
>    As shown in the table above, Cycle-GAN achieved an $R^2$ above 50% when the target ratio was set similarly to the original paper ($r=0.6/0.75$). However, when $r$ was reduced to below 0.1, the average $R^2$ of Cycle-GAN was degraded to below 10%. Therefore, our paper aims to enhance alignment for few trials using a flow-based framework.

---

> ### Author Response · Authors · 2024-11-22
> **Response to Weaknesses(3) of Reviewer EhkK**
>
> 3. **It is inappropriate to cherry-pick results in the presentation. Main Table 1 and Supplementary Table 8 are nearly identical. However, the authors included nine sessions per monkey in Table 1 but omitted two sessions that had the worst performances (Day 22 and 25 in CO-M; Day 40 and 77 in RT-M). Highlighting the best two or three sessions is one thing, but removing the worst two or three sessions is something entirely different.**
>
>    Thank you for pointing out this issue. Due to page limitations, we omitted the two sessions from CO-M and RT-M in Main Table 1. However, Supplementary Table 8 in the previous manuscript provided the complete results, including Day 22 and 25 for CO-M and Day 40 and 77 for RT-M. We sincerely apologize for this oversight in presentation.
>
>    The results for all sessions from CO-M and RT-M (including Day 22 and Day 25 in CO-M; Day 40 and Day 77 in RT-M) are now included in the updated Main Table 1. Additionally, the complete results for CO-C, which exhibit clearer distinctions, are further provided in the current Supplementary Table S5.

---

> ### Author Response · Authors · 2024-11-22
> **Response to Question1 of Reviewer EhkK**
>
> ### Questions:
>
> 1. **I am still confused about the term "attractor-like." The authors mentioned it in the context of "utilizing attractor-like ensemble dynamics (Gonzalez et al., 2019), a representation mechanism for encoding stimuli in the brain." I reviewed the referenced paper, but found that "attractor" is only mentioned once: "Second, attractor-like mechanisms ensure the persistence of representations over short periods of time (days), even if the animals are not exposed to the task or if the circuit is perturbed by lesions." I believe additional explanation of what you mean by "attractor-like" in this context would be helpful.**
>
>    Thank you for pointing out the unclear points. We explain what "attractor-like" means in this context and its relationship with our FDA in the following two points.
>
>    - **Meaning of “attractor-like”:** To better clarify the meaning of "attractor-like", we added more explanations and related references (Khona & Fiete, 2022; Gonzalez et al., 2019; Inagaki et al., 2019; Finkelstein et al., 2021; Hira et al., 2013). Among these references, the review paper (Khona & Fiete, Attractor and integrator networks in the brain. Nature Reviews Neuroscience, 2022, 23(12): 744-766.) reviewed research processes of attractors in neuroscience, which mainly explained the attractors as follows: “Despite the stochastic shifts within neural signals, certain shared low-dimensional neural manifolds exist in brain regions when similar tasks are performed. These manifolds often exhibit latent states that converge toward similar ones over time, a phenomenon known as attractor-like ensemble dynamics.” This mechanism inspires our use of attractor-like dynamics to extract consistent neural embeddings based on these convergent states, enabling the rapid adaptation of shifted neural signals within the neural manifold.
>
>      In addition, we have added additional explanations of attractor-like dynamics in the Introduction, including the following new sentences (Lines 69-76 on Page 2) and a new figure (Figure 1):
>
>      "_Despite the stochastic variability within neural recordings, regions like the motor cortex exhibit a shared low-dimensional manifold when similar tasks are performed. Within this manifold, latent states converge toward similar ones over time, a property known as attractor-like ensemble dynamics. This mechanism inspires us to leverage attractor-like ensemble dynamics, where the final similar states serve as neural embeddings. As shown in Figure 1, this dynamical property enables the rapid adaptation of raw neural signals with stochastic variability, thereby achieving consistent neural embeddings within the neural manifold._"
>
>     - **Relation with our FDA:** Building on the fact that attractor-like ensemble dynamics is a key property of dynamically stable systems (Bhatia N P, Szegö G P. Stability theory of dynamical systems. Springer Science & Business Media, 2002.), we propose FDA to establish such systems and achieve attractor-like dynamics. Specifically, our FDA approach utilizes flow matching to construct this dynamical system, with its stability theoretically verified based on incrementally input-to-state stability (Angeli D. A Lyapunov approach to incremental stability properties. IEEE Transactions on Automatic Control, 2002, 47(3): 410-421). The dynamical stability verification mainly relies on two key factors as detailed in the updated "Dynamical Stability Verification" section (Lines 250-256 on Page 5):
>
>       "_The dynamical stability is ensured by two key factors. First, the velocity field in flow matching is constructed using MLPs with Lipschitz-continuous activation functions. These functions ensure that latent state deviations remain stable under external input constraints, as shown in Eq.(7) and Eq.(21).  Second, the scale coefficient $\gamma^{S}$ of latent states is regularized to keep the ratio of latent state deviations between successive time steps below 1. This results in a geometric sequence with a ratio less than 1, causing latent states to gradually converge to similar ones, as presented in Eq.(6) and Eq.(22)._"
>
>       We have now highlighted this relationship in the Introduction (Lines 77-84 on Page 2) with the following newly added sentences:
>
>       "_In this work, based on the fact that attractor-like ensemble dynamic is a key property of dynamically stable systems, we propose a novel Flow-Based Dynamical Alignment (FDA) framework to establish such systems with attractor-like dynamics and achieve consistent neural embeddings. Specifically, our FDA approach leverages recent advances in flow matching, with the explicit likelihood maximization formulation provided by flows further facilitating a new source-free unsupervised alignment. The consistency of FDA embeddings was theoretically verified through the dynamical stability of neural manifolds, allowing for rapid adaptation with few target trials._"

---

> ### Author Response · Authors · 2024-11-22
> **Response to Question2 of Reviewer EhkK**
>
> ### Questions
>
> 2. **Azabou et al., 2023a and Azabou et al., 2023b are the same paper.**
>
>    Thank you for pointing out this issue. We have removed the redundant reference in the revised version.
>
>
> Thanks for the valuable comments and insightful suggestions, which have improved the clarity and rigor of our study. We hope that our responses and revisions have adequately addressed your concerns. We present a novel Flow-Based Dynamical Alignment (FDA) framework that utilizes attractor-like ensemble dynamics on stable neural manifolds. The FDA framework achieves consistent latent embeddings, as verified theoretically and experimentally. Our FDA provides a new approach for few-trial neural alignment, offering a new pathway to improve the chronic reliability of real-world BCIs.
>
> Therefore, we believe that our novel FDA framework will be of significant interest to the ICLR community, given its potential impact on few-trial neural alignment and real-world BCI reliability. Could you please consider raising the scores? We look forward to your valuable feedback. Thanks for your time and consideration.

---

> ### Author Response · Authors · 2024-11-25
>
> Thanks a lot for your valuable feedback. We have thoroughly gone through your comments and made revisions accordingly in the current manuscript. Specifically, we have added further explanations regarding the performance differences between NoMAD and Cycle-GAN, as well as additional clarifications on the “attractor-like dynamics.”
>
> We hope that these may address your concerns. We believe that our novel FDA framework will be of significant interest to the ICLR community, given its potential impact on few-trial neural alignment and real-world BCI reliability. Could you please consider raising the scores? We look forward to your further feedback. Thank you in advance.

---

> > ### Comment · Reviewer_EhkK · 2024-11-25
> > **Score and Data Presentation**
> >
> > I appreciate the detailed clarification from the authors regarding the discrepancy in results between FDA vs. NoMAD and Cycle-GAN, the lower performance of Cycle-GAN in this paper, and the attractor dynamics. This explanation makes the paper borderline (score 5 or 6) in terms of quality.
> >
> > I am, however, hesitant to raise my score to a 6, given that the authors removed two rows from their results. If the reason cited is "due to page limitations," then why were the two worst rows purposefully removed instead of selecting rows randomly or simply removing the last two? Reviewer EH95 also raised a similar concern, asking, "Figure 2d: why are there 9 days in Table 1 but 11 rows/columns in the shown matrices?" Personally, I strongly dislike this kind of selective presentation of experimental results.
> >
> > That said, I believe this work does make a valuable contribution to neural alignment in BCI, and I recommend it for acceptance (score 6).

---

> > > ### Author Response · Authors · 2024-11-26
> > >
> > > Thanks a lot for considering our responses and revising the score. We are glad to address any further concerns regarding the work.

---

### Official Review · Reviewer_RPD1 · 2024-10-31

**Soundness:** 3
**Presentation:** 3
**Contribution:** 3
**Rating:** 6
**Confidence:** 3

**Summary:**

This paper proposed a novel Flow-Based Dynamical Alignment framework to obtain consistent neural representation. The FDA approach uses flow matching techniques to extract dynamics on stable manifolds. The FDA work addressing the challenge of dynamical instability offers insights into neural dynamical stability. The latent extracted from FDA has better decoding performance.

**Strengths:**

1.	The proposed FDA method does perform better than baselines on decoding cursor velocity, and the learned latent space has more stability than baselines.
2.	The proposed method has relatively better computational efficiency although may need more parameters.
3.	The authors did multiple ablation studies on main components.

**Weaknesses:**

One weakness (as is already notified by the authors) is the lack of cross-subject validation, for example, between Monkey C and M.

**Questions:**

1.	The results on Monkey C seem very different from Monkey M, could you explain the reason?
2.	How many trials are in the source domain?
3.	The authors computed average MLE, what is the average across? Then what are all MLEs instead of the average one?
4.	Is fine tuning the reason that FDA performs better on few-trials scenarios?
5.	How do you choose the hyper parameters? Especially the dimensionality of your embedded latent space. Also, when you compare all the different models, do they have the same latent dimension?
6.	I am just curious, in Table 1, for each method, the worst r2 is in a different day, e.g., in CO-M, LSTM has the worst r2 in day29, but in FDA-MLA, it is just day8. Could you explain the reason?

---

> ### Author Response · Authors · 2024-11-22
> **Response(1) to Reviewer RPD1**
>
> Thank you for the thorough read of our manuscript and insightful suggestions. We provided a point-by-point response to your comments and suggestions below and revised the manuscript accordingly.
>
> ### Weaknesses:
>
> 1. **One weakness (as is already notified by the authors) is the lack of cross-subject validation, for example, between Monkey C and M.**
>
>    Thanks for the suggestion. However, unlike EEG signals which are recorded with standard placement of electrodes, spike signals of different subjects can be highly different due to the inconsistent placement and non-stationary neuronal activities. Therefore, it is usually difficult to perform cross-subject evaluation for such signals. We agree that the cross-subject validation for spike signals can be an interesting topic for future studies.
>
> ### Questions:
>
> 1. **The results on Monkey C seem very different from Monkey M, could you explain the reason?**
>
>    Thank you for raising this valuable question. We think that this difference was caused by the inherent instability of neural signals from CO-C, which could have a substantial influence under few-trial scenarios. To further investigate this, we evaluated the performance of FDA-MMD on both Monkey C and Monkey M under varying target ratios.
>
>    As shown in the table below, although FDA-MMD performed not good enough on CO-C with low target ratios, its performance improved and became comparable to that of Monkey M when the target ratio exceeded 0.3 (around 60 trials). The corresponding explanations are added to Appendix C.1.2 and Table S6 on Page 21 in the present manuscript.
>
>    **Comparison of average $R^2$ values (\%) across sessions for FDA-MMD on the CO-C, CO-M, and RT-M datasets ($r = 0.02$). The average standard deviations over five runs per session are also reported.**
>
>    | $r$ | 0.02 | 0.03 | 0.04 | 0.06 | 0.1 | 0.2 | 0.3 | 0.4 | 0.5 | 0.6 |
>    |:---------:|:------:|:------:|:------:|:------:|:-----:|:-----:|:-----:|:-----:|:-----:|:-----:|
>    | CO-C | 16.40 ± 5.40 | 17.08 ± 7.53 | 17.27 ± 8.58 | 17.41 ± 7.66 | 28.18 ± 5.36 | 42.61 ± 5.23 | 50.12 ± 6.90 | 54.87 ± 5.05 | 55.05 ± 5.71 | 56.00 ± 4.88 |
>    | CO-M | 45.59 ± 5.15 | 48.40 ± 4.59 | 50.71 ± 4.68 | 51.10 ± 4.76 | 57.90 ± 2.68 | 62.20 ± 2.41 | 65.16 ± 2.53 | 66.38 ± 2.44 | 66.78 ± 2.48 | 67.32 ± 3.32 |
>    | RT-M | 42.08 ± 6.31 | 44.36 ± 5.83 | 45.35 ± 6.15 | 47.23 ± 5.96 | 52.15 ± 4.16 | 53.66 ± 3.35 | 55.28 ± 2.89 | 56.45 ± 2.89 | 56.53 ± 2.55 | 57.93 ± 2.39 |
>
> 2. **How many trials are in the source domain?**
>
>    The source domain contains approximately 200 trials, and we have added this information to the 'Data Preprocessing and Split' (Line 366 on Page 7, Section 4.1) in the present manuscript. Thanks a lot.
>
> 3. **The authors computed average MLE, what is the average across? Then what are all MLEs instead of the average one?**
>
>    The average MLE is computed across five random runs of pre-training for each individual source session. We have updated this information in the legend of Figure 3(a) on Page 9 in the present manuscript.
>
>    To investigate all MLEs, we further visualized the distribution of MLEs across target sessions using violin plots. As shown in Figure S2(a), in contrast to ERDiff and NoMAD, FDA achieved negative MLEs in most cases, which aligns with the average MLE results. This demonstrates the dynamical stability of our pre-trained neural manifolds. Additional details are provided in Appendix C.1.1 and Figure S2(a) on Page 20 in the present manuscript. Thanks a lot.

---

> ### Author Response · Authors · 2024-11-22
> **Response(2) to Reviewer RPD1**
>
> ### Questions
>
> 4. **Is fine tuning the reason that FDA performs better on few-trials scenarios?**
>
>    Thank you for this valuable comment. Fine-tuning did contribute to the improved performance in few-trial scenarios. To verify this, we compared the cross-session performance of NoMAD without alignment, Cycle-GAN without alignment, and FDA without alignment on the CO-M and RT-M datasets.
>
>    As shown in the table below, we observed that FDA outperformed the baselines without alignment, due to the dynamical stability of its pre-trained latent spaces. Furthermore, the performance of FDA in few-trial scenarios continued to improve after fine-tuning. Thus, we conclude that fine-tuning is the reason for FDA's superior performance in few-trial scenarios. Related content is available in Appendix C.1.4 and Table S9 on Page 22 in the present manuscript.
>
>    **Comparison of $R^2$ values (in \%) across target sessions (where the $R^2$ scores for each session are averaged over five random runs with different sample selections) of baselines and FDA without alignment on CO-M and RT-M datasets**
>
>    | Data | NoMAD w/o alignment | Cycle-GAN w/o alignment | FDA w/o alignment | FDA-MLA | FDA-MMD |
>    |:-------:|:--------------------:|:-----------------------:|:-------------------:|:---------:|:---------:|
>    | CO-M  | -121.47 ± 77.80     | -126.84 ± 23.82       | 16.23 ± 9.43       | 36.05 ± 5.84 | 45.59 ± 5.15 |
>    | RT-M  | -74.06 ± 49.94      | -3.42 ± 5.55          | 38.15 ± 8.21       | 41.73 ± 4.88 | 42.08 ± 6.31 |
>
> 5. **How do you choose the hyper parameters? Especially the dimensionality of your embedded latent space. Also, when you compare all the different models, do they have the same latent dimension?**
>
>    Thank you for pointing out the unclear points. The dimensionality of embedded latent spaces was selected primarily based on their cross-session performance, determined through grid searches. For a fair comparison, different models were evaluated using their respective best latent dimensions.
>
>    Specifically, we conducted grid-search experiments on the latent dimensions of NoMAD and CEBRA. As shown in the tables below, we selected the latent dimensions for NoMAD and CEBRA as 16 and 32, respectively. For ERDiff, we set the latent dimension to 8, following the default settings specified in the original paper, as it was applied to similar datasets. The corresponding details are provided in Appendix C.1.3, Table S7, and Table S8 on Page 21 and 22 in the present manuscript.
>
>    **Average $R^2$ scores across target sessions of NoMAD on CO-M and RT-M datasets under different latent dimensions**
>    | Latent Dimension | 12            | 16            | 32            | 48            |
>    |:------:|:---------------:|:---------------:|:---------------:|:---------------:|
>    | CO-M | 4.97 ± 8.29 | **6.40** ± 6.22 | 3.69 ± 7.00 | -6.21 ± 8.70 |
>    | RT-M | 3.42 ± 8.78 | **11.74** ± 6.42 | 8.27 ± 10.02 | 2.42 ± 9.21 |
>
>    **Average $R^2$ scores across target sessions of CEBRA on CO-M and RT-M datasets under different latent dimensions**
>    | Latent Dimension | 16            | 32            | 48            |
>    |:-----------------:|:--------------:|:--------------:|:--------------:|
>    | CO-M | -1.34 ± 11.69 | **1.14** ± 14.47 | 0.85 ± 12.61 |
>    | RT-M | -53.01 ± 14.49 | **-45.48** ± 12.51 | -49.21 ± 14.71 |
>
> 6. **I am just curious, in Table 1, for each method, the worst r2 is in a different day, e.g., in CO-M, LSTM has the worst r2 in day29, but in FDA-MLA, it is just day8. Could you explain the reason?**
>
>    Thank you for raising this interesting point. We think that this variability stems from the different criteria used to search for the optimal alignment. To illustrate this, we analyzed the negative log likelihood (NLL) curve of FDA-MLA on Day 8 (CO-M) as an example.
>
>    As shown in Figure S2(b), FDA-MLA exhibited an abnormal increase in NLL during the initial fine-tuning epochs. In contrast, other methods, such as NoMAD (based on KL divergences) and LSTM (without alignment), did not exhibit this phenomenon on the same day. Additional details are provided in Appendix C.1.2 and Figure S2(b) on Page 20 in the present manuscript.
>
> Thank you again for the constructive feedback, which we believe to help improve the clarity and rigor of our study. We hope that our responses and revisions have adequately addressed your concerns. In this work, we present a novel Flow-Based Dynamical Alignment (FDA) framework that leverages attractor-like ensemble dynamics to provide a new approach for few-trial neural alignment. Therefore, we believe that our novel FDA framework will be of significant interest to the ICLR community.
>
> Could you please consider raising the scores? We look forward to your valuable feedback. Thanks for your time and consideration.

---

> > ### Comment · Reviewer_RPD1 · 2024-11-22
> >
> > I would like to thank the authors for their responses, and I appreciate all your efforts in addressing my concern. After reading your updated manuscript and other reviewers’ concerns, I have new questions. (1) Is the model only tested on few trials of large data? If so, how could you select these trials? (2) The proposed model has similar performance to baselines with large numbers of targets, but outperforms baselines with fewer targets, right? Are they tested on the same targets? (3) In your MLE plot, for the same source, are MLEs in 5 random runs different? This is important, if some of them are negative but others have positive value, we may not be able to say it is stable.

---

> > > ### Author Response · Authors · 2024-11-23
> > >
> > > Thanks for your reply and the additional feedback. We hope the following responses address your concerns effectively.
> > >
> > > **(1)	Is the model only tested on few trials of large data? If so, how could you select these trials?**
> > >
> > > Our model is fine-tuned using a few trials (5 out of 200) selected randomly and tested on all remaining trials after the fine-tuning phase. Specifically, pre-training was conducted 5 times with random initializations, with each pretrained model fine-tuned on 5 distinct random selections of trials distinct from those used with other pretrained models. This yields tests across 25 random few-trial selections per session, which we believe is sufficient to rule out accidental superior performances.
> > >
> > > **(2)	The proposed model has similar performance to baselines with large numbers of targets, but outperforms baselines with fewer targets, right? Are they tested on the same targets?**
> > >
> > > Our FDA-MMD (CO-M: 67.32%, RT-M: 57.93%) still outperformed other baselines, including Cycle-GAN (CO-M: 62.46%, RT-M: 49.41%) and NoMAD (CO-M: 50.90%, RT-M: 56.10%), even as target ratios increased (e.g., $r$=0.6), though the performance gap narrowed. Given that FDA-MLA is source-free, unlike the chosen baselines, its performance with larger target ratios (CO-M: 56.61%, RT-M: 46.63%) is acceptable. In addition, the same target trials were used for alignment across FDA and other baselines.
> > >
> > > **(3)	In your MLE plot, for the same source, are MLEs in 5 random runs different? This is important, if some of them are negative but others have positive value, we may not be able to say it is stable.**
> > >
> > > The MLEs across 5 random runs vary due to differences in initialization for the same source. However, we observed that nearly all MLEs achieved by FDA are non-positive (CO-M: 55/55, RT-M: 52/55), with a non-positive MLE generally indicating dynamical stability, as discussed in Line 411 on Page 8. The three exceptions have MLEs below 1e-3, which can be considered approximately stable. Therefore, we conclude that the system is stable.
> > >
> > > Thank you once again for your time and thoughtful consideration. We look forward to your further feedback.

---

### Official Review · Reviewer_EH95 · 2024-11-04

**Soundness:** 3
**Presentation:** 3
**Contribution:** 3
**Rating:** 6
**Confidence:** 3

**Summary:**

The paper proposes FDA - an alignment method to maintain performance of BCI decoder across recording sessions. The method is based on flow matching to achieve consistent neural embeddings which is theoretically shown to be dynamically stable and facilitates alignment in few-trial scenarios. Alignment performance is validated on multi-day datasets of monkeys performing motor tasks, showing competitive results against other baselines.

**Strengths:**

* The method based on flow matching is original, tackling an important and long standing issue in BCI field.
* Problem formulation makes sense. Quantitative results are quite thorough with comparison using several baselines and datasets.
* The paper is well written for the most part. Figures are clear and well presented.

**Weaknesses:**

* The paper cited a recent related work [1] but did not compare with it. Incorporating this additional baseline would provide a more comprehensive evaluation of the proposed method.
* Some texts are unclear and need more elaboration (details in Questions).

[1] Ayesha Vermani, Il Memming Park, and Josue Nassar. Leveraging generative models for unsupervised alignment of neural time series data. In The Twelfth International Conference on Learning Representations, 2024.

**Questions:**

* What aspect in the proposed method uses "attractor-like ensemble dynamics”? Can the authors provide more clarification on what “attractor-like ensemble dynamics” mean and how it is relevant to certain part of FDA? The term is used a lot in the Introduction but was never referred to again in the Methodology or Experiments & Results.
* Line 124: how was the signal window sampled? how many windows are sampled per trial? Are the windows overlap with each other?
* Line 125: why behavior label $y_i$ is only taken at the w-th timestep of $x_i$? Does this mean the method decodes the downsampled behavior time series rather than the original one? The other baselines didn’t seem to have the behavior target downsampled. Decoding the downsampled behavior might make it an easier task than decoding the original behavior.
* Figure 1: in the first block, shouldn’t $c^S$ be at the top and $c^T$ be at the bottom?
* Also figure 1: according to description in Methodology section, $x^T$ and $z^T$ should not be used during Pre-training phase (left and middle blocks)?
* Line 185: does the transformer utilize positional embeddings? If so, what kind of positional embeddings was used?
* Line 206: how was $\eta$ pre-defined? Is $\eta$ kept the same across days?
* Figure 2c: Is each point on the plot the average of all test sessions or average of different choices of samples with the same ratio $r$? Providing this clarification and also adding errorbars for each point would make it more informative.
* Also figure 2c: will performance of other baselines improve and reach the same performance of FDA if $r$ increases? If so, at how many trials will they become comparable to FDA? This is important to gauge the helpfulness of FDA in cases where scarcity of target samples is not a problem.
* Figure 2d: why there are 9 days in Table 1 but 11 rows/columns in the shown matrices?

---

> ### Author Response · Authors · 2024-11-22
> **Response(1) to Reviewer EH95**
>
> Thank you for the thorough read of our manuscript and insightful suggestions. We provided a point-by-point response to your comments and suggestions below and revised the manuscript accordingly.
>
> ### Weaknesses:
>
> 1. **The paper cited a recent related work [1] but did not compare with it. Incorporating this additional baseline would provide a more comprehensive evaluation of the proposed method.**
>
>    Thank you for pointing out this issue. We failed to find public codes for this work [1]. However, owing to the similar architecture (seq-VAE) and comparable performance on non-human primate datasets reported in Table 2 of [1], we used NoMAD as a baseline instead. We will replicate the code of this work and compare with it in future work.
>
>    _[1] Ayesha Vermani, Il Memming Park, and Josue Nassar. Leveraging generative models for unsupervised alignment of neural time series data. In The Twelfth International Conference on Learning Representations, 2024._

---

> ### Author Response · Authors · 2024-11-22
> **Response(2) to Reviewer EH95**
>
> ### Questions:
>
> 1.  **What aspect in the proposed method uses "attractor-like ensemble dynamics”? Can the authors provide more clarification on what “attractor-like ensemble dynamics” mean and how it is relevant to certain part of FDA? The term is used a lot in the Introduction but was never referred to again in the Methodology or Experiments &Results.**
>
>     Thanks for pointing out the unclear points. We give more detailed explanations for "attractor-like" here and revise the article accordingly.
>
>     - **About the meaning of "attractor-like":** To better clarify the meaning of "attractor-like, we added more explanations and related references [2-4]. The meaning of "attractor-like" is explained as follows. Despite the stochastic variability in neural signals, the shared low-dimensional neural manifolds [2] exist in brain regions when similar tasks are performed. These manifolds often exhibit latent states converging toward stable and similar ones over time, a phenomenon known as attractor-like ensemble dynamics.  This property motivates us to establish attractor-like dynamics for consistent neural embeddings based on these convergent states, facilitating the rapid adaptation of shifted neural signals within the neural manifold.
>
>       In addition, we have added additional explanations of attractor-like dynamics in the Introduction, including the following new sentences (Lines 69-76 on Page 2) and a new figure (Figure 1):
>
>       _"Despite the stochastic variability within neural recordings, regions like the motor cortex exhibit a shared low-dimensional manifold when similar tasks are performed. Within this manifold, latent states converge toward similar ones over time, a property known as attractor-like ensemble dynamics. This mechanism inspires us to leverage attractor-like ensemble dynamics, where the final similar states serve as neural embeddings. As shown in Figure 1, this dynamical property enables the rapid adaptation of raw neural signals with stochastic variability, thereby achieving consistent neural embeddings within the neural manifold."_
>
>     - **Relation with our FDA:** According to the fact that attractor-like ensemble dynamics is a typical property of dynamically stable systems [3], we propose FDA to establish such systems and achieve attractor-like dynamics. Specifically, our FDA framework leverages flow matching to implement this dynamical system, with its incrementally input-to-state stability [4] theoretically ensured by Lipschitz-continuous activation functions and regularized scale coefficients, as detailed in the updated "Dynamical Stability Verification" section (Lines 250-256 on Page 5).
>
>       We have now highlighted this relationship in the Introduction (Lines 77-84 on Page 2) with the following newly added sentences:
>
>       _"In this work, based on the fact that attractor-like ensemble dynamics is a key property of dynamically stable systems, we propose a novel Flow-Based Dynamical Alignment (FDA) framework to establish such systems with attractor-like dynamics and achieve consistent neural embeddings. Specifically, our FDA approach leverages recent advances in flow matching, with the explicit likelihood maximization formulation provided by flows further facilitating a new source-free unsupervised alignment. The consistency of FDA embeddings was theoretically verified through the dynamical stability of neural manifolds, allowing for rapid adaptation with few target trials."_
>
>     In the original manuscript,  we frequently used the term 'dynamical stability' instead of 'attractor-like ensemble dynamics' in the Methodology and Experiments & Results sections. To alleviate this gap, we have added transition statements at the beginning (Lines 155-158) of Section 3.2, including the following new sentences:
>
>     _"To obtain consistent neural embeddings from non-stationary neural signals, we propose a novel framework that applies flow matching on neural manifolds, constructing a dynamically stable system to achieve attractor-like ensemble dynamics."_
>
>     _[2] Khona M, Fiete I R. Attractor and integrator networks in the brain. Nature Reviews Neuroscience, 2022, 23(12): 744-766._
>
>     _[3] Bhatia N P, Szegö G P. Stability theory of dynamical systems. Springer Science & Business Media, 2002._
>
>     _[4] Angeli D. A Lyapunov approach to incremental stability properties. IEEE Transactions on Automatic Control, 2002, 47(3): 410-421._

---

> ### Author Response · Authors · 2024-11-22
> **Response(3) to Reviewer EH95**
>
> ### Questions:
>
> 2. **Line 124: how was the signal window sampled?  how many windows are sampled per trial? Are the windows overlap with each other?**
>
>    Thank you for pointing out these unclear points, which we should be clearer. There are approximately 20 sampled windows per trial, with each window overlapping the previous one. Specifically, the first signal window of a trial is sampled from the first time point to the $w$-th time point. The second window starts from the second time point, one step behind the first one. Additional details have been included in Section 3.1 (Lines 140-142 on Page 3) with the following sentence:
>
>    _"The first signal window of each trial begins at the initial time point, while the second window starts one step later."_
>
> 3. **Line 125: why behavior label is only taken at the w-th timestep of $x_i$? Does this mean the method decodes thedownsampled behavior time series rather than the original one? The other baselines didn't seem to have the behavior target downsampled. Decoding the downsampled behavior might make it an easier task than decoding the original behavior.**
>
>    Thanks for raising these questions. No downsampling has been performed on behavior labels. The behavior label is only assigned at the $w$-th timestep for two main reasons. First, we believe that short-time causal windows are better suited for real-time decoding than direct decoding of the entire trial. Second, utilizing the $w$ previous points as context information is expected to improve the decoding performance. To clarify this further, we have added the following sentence to Section 3.1 (Lines 142-145 on Page 3):
>
>    _"The behavioral label is assigned at the $w$-th time step to meet real-time decoding requirements using short-time causal windows and to leverage previous time steps as contextual information effectively."_
>
> 4. **Figure 1: in the first block, $c^S$ shouldn't be at the top and $c^T$ be at the bottom? Also figure 1: according to description in Methodology section, $x^T$ and $z^T$ should not be used during Pre-training phase (left and middle blocks)?**
>
>    Thank you for pointing out these issues in Figure 1. We have swapped the positions of $c^S$ and $c^T$, and included $x^T$ and $z^T$ in the fine-tuning phase (right block). The revised illustration is now provided as Figure 2 on Page 4 in the present manuscript.
>
> 5. **Line 185: does the transformer utilize positional embeddings? If so, what kind of positional embeddings was used?**
>
>    Thanks for the question. The transformer utilizes the classical Sinusoidal Positional Encoding in our work. This information has been included in “Conditional Feature Extraction Based on Neural Dynamics” of Section 3.2.1 (Lines 203-205 on Page 4) in the present manuscript.
>
> 6. **Line 206: how was η pre-defined? Is η kept the same across days?**
>
>     Thanks for raising this lack of clarity. $\eta$ was pre-defined using Xavier initialization and was kept the same across days. Additional clarification has been included in "Flow Matching Conditioned on Latent Dynamics" of Section 3.2.1 (Lines 224-225 on Page 5) in the present manuscript.
>
> 7. **Figure 2c: Is each point on the plot the average of all test sessions or average of different choices of samples with the same ratio $r$? Providing this clarification and also adding errorbars for each point would make it more informative.**
>
>    Thank you for raising the valuable comments. Each point on the plot represents an average across all target sessions, as well as five random selections of target samples from each session. Clarifications and additional error bars have been added to Section 4.2.2 (Lines 432-434 on Page 9) and Figure 3(c) in the present manuscript.
>
> 8. **Also figure 2c: will performance of other baselines improve and reach the same performance of FDA if $r$ increases? If so, at how many trials will they become comparable to FDA? This is important to gauge the helpfulness of FDA in cases where scarcity of target samples is not a problem.**
>
>    Thank you for the insightful comments. The performance of other baselines improves and becomes comparable when r reaches approximately 0.3 (around 60 trials). FDA and other baselines provided comparative performance, where target samples are relatively sufficient. Detailed results are provided in Appendix C.1.5, including Figure S4 and the newly added sentences (Lines 1163-1168 on Page 22) in the present manuscript:
>
>    _"To further evaluate the performance of FDA under different target ratios $r$, we gradually increased $r$ from 0.02 to 0.6. The $R^2$ scores for NoMAD, Cycle-GAN, and FDA are shown in Figure S4. In particular, Cycle-GAN and NoMAD exhibited significantly lower performance (approximately five times worse) with fewer target samples. However, as r increased to around 0.3 (approximately 60 trials), their performance became comparable to that of FDA-MLA and FDA-MMD."_

---

> ### Author Response · Authors · 2024-11-22
> **Response(4) to Reviewer EH95**
>
> ### Questions:
>
> 9. **Figure 2d: why there are 9 days in Table 1 but 11 rows/columns in the shown matrices?**
>
>    Thank you for pointing out this issue, and we sincerely apologize for the confusion. The original Table 1 omitted two sessions from the CO-M and RT-M datasets due to space limitations. We have updated Table 1 on Page 8 in the present manuscript to include all 11 sessions.
>
> Thank you again for the constructive feedback, which we believe to help improve the clarity and rigor of our study. We hope that our responses and revisions have adequately addressed your concerns. In this work, we present a novel Flow-Based Dynamical Alignment (FDA) framework that leverages attractor-like ensemble dynamics to provide a new approach for few-trial neural alignment. Therefore, we believe that our novel FDA framework will be of significant interest to the ICLR community.
>
> Could you please consider raising the scores? We look forward to your valuable feedback. Thanks for your time and consideration.

---

> ### Author Response · Authors · 2024-11-25
>
> Thanks a lot for your valuable feedback. We have thoroughly gone through your comments and made revisions accordingly in the current manuscript.
>
> We hope that these responses and revisions may address your concerns. We believe that our novel FDA framework will be of significant interest to the ICLR community, given its potential impact on few-trial neural alignment and real-world BCI reliability. Could you please consider raising the scores? We look forward to your further feedback. Thank you in advance.

---

> > ### Comment · Reviewer_EH95 · 2024-12-03
> >
> > I thank the authors for addressing my questions. I would like to maintain my scores.

---

### Official Review · Reviewer_Fr7z · 2024-11-04

**Soundness:** 3
**Presentation:** 3
**Contribution:** 3
**Rating:** 6
**Confidence:** 2

**Summary:**

This paper proposes an consistent neural embeddings using flow matching, and leverages attractor-like ensemble dynamics. The numerical experiments showed consistent alignment results and better results than existing algorithms. This paper also theoretically showed the stability on the alignment using the algorithm.

**Strengths:**

This paper uses the source-free alignment via likelihood maximization (FDA-MLA) and uses pre-training and fine-tuning to achieve consistent neural embeddings from non-stationary neural signals. The set up of latent extraction is via conditional feature extraction based on neural dynamics.

**Weaknesses:**

1. The dynamical stability verification is not clearly explained for the theoretical support.

2. Some results are puzzling (see Questions below).

**Questions:**

1. In part 3.2.2, can you explain the reason of using maximum mean discrepancy, why it is better than other matrix, can you show it ?

2. For table 1, can you explain why the results of ERDiff is extremely different and worse than the others, that is a relatively new paper published in NeurIPS 2023.

3. For table 1, why the R2(%) decreases to 23.79(FDA-MLA) and 45.23(FDA-MMD) at Day 8, but increases to 50.15 and 55.9? In the Cycle--GAN paper, the R2 decreases continuously. My understanding is that the alignment should be worser with longer time-gap relative to Day 0.

4. For table 1, why the R2(%) of CEBRA is much better in RT-M dataset than CO-M dataset?

---

> ### Author Response · Authors · 2024-11-22
> **Response(1) to Reviewer Fr7z**
>
> Thank you for the thorough read of our manuscript and insightful suggestions. We revised the manuscript according to your comments and suggestions, and provided a point-by-point reply to your questions.
>
> ### Weaknesses:
>
> 1. **The dynamical stability verification is not clearly explained for the theoretical support.**
>
>    Thanks for raising this lack of clarity. The dynamical stability verification relies on two key factors: (1) the velocity field in flow matching is constructed utilizing MLPs with Lipschitz-continuous activation functions, consequently stabilizing latent state deviations under input constraints; and (2) the scale coefficient is regularized to maintain a geometric sequence with a ratio less than one, to promote the gradual convergence of latent state deviations.
>
>    A detailed explanation has been added to "Dynamical Stability Verification" of Section 3.2.1 (Lines 250-256 on Page 5), including the following new sentences:
>
>    _"The dynamical stability is ensured by two key factors. First, the velocity field in flow matching is constructed using MLPs with Lipschitz-continuous activation functions. These functions ensure that latent state deviations remain stable under external input constraints, as shown in Eq.(7) and Eq.(21).  Second, the scale coefficient $\gamma^S$ of latent states is regularized to keep the ratio of latent state deviations between successive time steps below 1. This results in a geometric sequence with a ratio less than 1, causing latent states to gradually converge to similar ones, as presented in Eq.(6) and Eq.(22)."_
>
> ### Questions:
>
> 1. **In part 3.2.2, can you explain the reason of using maximum mean discrepancy, why it is better than other matrix, can you show it?**
>
>     Thanks for the question. Due to the scarcity of target samples, criteria based on the probability density of individual samples, such as KL divergence in GANs, can result in gradient instability during fine-tuning. In contrast, Maximum Mean Discrepancy (MMD) utilizes higher-order moments as overall sample properties, effectively mitigating the impact of outliers in limited samples. This explanation has been added to "Maximum Mean Discrepancy Alignment with Few Target Trials" of Section 3.2.2 (Lines 293-297 on Page 6) with the following new sentences:
>
>     _"When target sizes are small, the alignment based on individual sample probabilities, such as Kullback-Leibler (KL) divergences in GANs, often leads to training instability. In contrast, Maximum Mean Discrepancy (MMD) leverages higher-order moments as overall sample properties, effectively reducing the influence of outliers in limited samples."_
>
>     To illustrate this, we compared alignment methods on the same variable using GANs (FDA-g) and MMD (FDA-MMD), as shown in Figure 4(a) and Figure S6(a). The $R^2$ curves on target sessions demonstrate the instability of GAN-based alignment (FDA-g) during the fine-tuning phase. In contrast, MMD-based alignment (FDA-MMD) exhibits significantly more stable curves, demonstrating its robustness to outliers in few-trial scenarios. We have shown the better alignment based on MMD in Figure 4(a) and Figure S6(a) on Page 10 and Page 24, respectively.
>
> 2. **For table 1, can you explain why the results of ERDiff is extremely different and worse than the others, that is a relatively new paper published in NeurIPS 2023.**
>
>    Thank you for pointing out this issue. We used the original code from the authors, but encountered vanishing gradient problems when applying it to our datasets. Upon investigation, we found this issue may be related to the calculation of Sinkhorn Divergences. We refined the original calculation method to address this problem,  and obtained the results reported in our paper. Additionally, our results were similar to those reported in Table 2 ($R^2$=-0.32) of [1]. Related content has been added to Section 4.2.2 (Lines 426-427 on Page 8) as follows:
>
>    _"ERDiff often showed negative scores, aligning with results reported in (Vermani et al., 2024)."_
>
>    _[1] Vermani A, Park I M, Nassar J. Leveraging Generative Models for Unsupervised Alignment of Neural Time Series Data. In The Twelfth International Conference on Learning Representations, 2024._

---

> ### Author Response · Authors · 2024-11-22
> **Response(2) to Reviewer Fr7z**
>
> ### Questions:
>
> 3. **For table 1, why the R2(%) decreases to 23.79(FDA-MLA) and 45.23(FDA-MMD) at Day 8, but increases to 50.15 and 55.9? In the Cycle--GAN paper, the R2 decreases continuously. My understanding is that the alignment should be worser with longer time-gap relative to Day 0.**
>
>    Thanks for this valuable question. We found that this fluctuation in $R^2$ was caused by the instability of specific outlierscertain samples. Since the target ratio was small, the impact of these samples can be significant.
>
>    As a further analysis for this question, we conducted experiments using Cycle-GAN and FDA-MMD under varying target ratios (0.2, 0.4, and 0.6) on the CO-M dataset. As shown in  Figure S5, both FDA-MMD and Cycle-GAN displayed fluctuating $R^2$ curves at smaller target ratios.  Notably, the decrease on certain days, such as Day 8, Day 22, and Day 25, suggests that the model may be affected by specific outliers, despite the shorter time-gap relative to Day 0. Notably, the decrease on certain days, such as Day 8, suggests that the model may be affected by outliers within the limited target samples, despite the shorter time gap relative to Day 0. However, as the target ratio increased, the fluctuation degraded. When the target ratio reached 0.6, the $R^2$ mostly decreased continuously across sessions, consistent with the trends reported in the original Cycle-GAN paper. To clarify these points, we have added Figure S5 of Appendix C.1.5 in the present manuscript.
>
> 4. **For table 1, why the R2(%) of CEBRA is much better in RT-M dataset than CO-M dataset?**
>
>    Thank you for this question. The better performance of CEBRA, a method without alignment, may stem from inherently smaller gaps between sessions in RT-M. To further investigate this, we analyzed the performance of NoMAD, Cycle-GAN, and FDA without alignment on both CO-M and RT-M datasets.
>
>    As shown in the table below, all methods without alignment achieved significantly better cross-session performance on RT-M than CO-M dataset. In contrast, FDA-MMD and FDA-MLA, which are methods incorporating alignment, both demonstrated comparative performance on the CO-M and RT-M datasets.
>
>     **Comparison of $R^2$ values (in \%) across target sessions (where the $R^2$ scores for each session are averaged over five random runs with different sample selections) of baselines and FDA without alignment on CO-M and RT-M datasets**
>
>    | Data | NoMAD w/o alignment | Cycle-GAN w/o alignment | FDA w/o alignment | FDA-MLA | FDA-MMD |
>    |:-------:|:--------------------:|:-----------------------:|:-------------------:|:---------:|:---------:|
>    | CO-M  | -121.47 ± 77.80     | -126.84 ± 23.82       | 16.23 ± 9.43       | 36.05 ± 5.84 | 45.59 ± 5.15 |
>    | RT-M  | -74.06 ± 49.94      | -3.42 ± 5.55          | 38.15 ± 8.21       | 41.73 ± 4.88 | 42.08 ± 6.31 |
>
>    Relevant results were added to Appendix C.1.4.
>
> Thank you again for the constructive feedback, which we believe to help improve the clarity and rigor of our study. We hope that our responses and revisions have adequately addressed your concerns. In this work, we present a novel Flow-Based Dynamical Alignment (FDA) framework that leverages attractor-like ensemble dynamics to provide a new approach for few-trial neural alignment. Therefore, we believe that our novel FDA framework will be of significant interest to the ICLR community.
>
> Could you please consider raising the scores? We look forward to your valuable feedback. Thanks for your time and consideration.

---

> ### Author Response · Authors · 2024-11-25
>
> Thanks a lot for your valuable feedback. We have thoroughly gone through your comments and made revisions accordingly in the current manuscript.
>
> We hope that these responses and revisions may address your concerns. We believe that our novel FDA framework will be of significant interest to the ICLR community, given its potential impact on few-trial neural alignment and real-world BCI reliability. Could you please consider raising the scores? We look forward to your further feedback. Thank you in advance.

---

> > ### Comment · Reviewer_Fr7z · 2024-12-02
> >
> > I agree that this paper will be interested to the ICLR community and your modification and feedback clearly solved my concerns. I don't have any further questions. Thanks!

---

### Meta-Review · Area_Chair_cmx5 · 2024-12-18

**Metareview:**

This paper proposes Flow-Based Dynamical Alignment (FDA), a method for aligning neural embeddings across recording sessions using flow matching techniques. The approach aims to address the challenge of dynamical instability and achieve consistent neural representations, validated through experiments that demonstrate improved decoding performance and alignment compared to several existing methods.

The paper's strengths lie in its innovative use of flow matching, which tackles a critical issue in brain-computer interface research. The authors conducted extensive benchmarking against multiple baselines, showing competitive improvements in decoding performance and stability. The methodology is computationally efficient and includes ablation studies to validate key components of the proposed framework.

However, several weaknesses detract from the paper's overall impact, as outlined below. I appreciate the reviewers' comments, which have significantly helped improve the paper, as well as the authors' rebuttal. However, I believe there are some fundamental issues, as mentioned below, that cannot be fully addressed within the brief rebuttal period. I recommend that the authors continue refining the paper to enhance its quality for future submissions.

**Additional Comments On Reviewer Discussion:**

There appears to be a fundamental disconnect between the model's motivation and its modeling itself. The paper claims that existing methods fail because they cannot find a "consistent neural manifold." However, this claim is vague—what does "consistent" mean in this context? Isn't the purpose of neural alignment methods precisely to address inconsistencies in neural embeddings? The paper also states that existing representation techniques may yield inconsistent neural embeddings due to stochastic perturbations in neural recordings. But if there were no stochastic perturbations, neural alignment wouldn't be necessary in the first place.

Moreover, the cited papers are indeed learning neural dynamics, as they adopt a VAE/generative framework where the latent variable z can reconstruct neural data, albeit influenced by behavior decoding. In contrast, the proposed method does not offer a generative model for neural signals; its latent variable z is tied to behavioral labels (y) rather than neural data (x). Consequently, it is misleading to claim that the discovered latent variable reflects a neural manifold or neural dynamics. This disconnect is further highlighted by the initial assumption about attractor-like embeddings. The paper uses neural dynamics to motivates but I don't think their z is the latent dynamic of neural data, no interpretation or visualization. Also the claim about attractor like embedingb lacks strong support from the cited studies. The referenced works focus on hippocampal dynamics or areas such as the mouse premotor cortex but do not provide robust evidence for attractor-like dynamics in the monkey's primary motor cortex (M1). Even if such dynamics existed, the connection between the proposed ODE framework and attractor-like behavior is tenuous. While most neural latent dynamics models rely on discrete state-space models (a discrete analog of ODEs), some, like Kim et al. (ICML 2021), directly assume continuous ODEs. If continuous dynamics are central to the method's benefits, the motivation should be more explicit. Instead, the authors justify using continuous normalizing flows by critiquing discrete flows for their constrained representation capacity, further muddying the narrative. The attractor dynamics motivation disappears after the introduction and finds no support in the experiments.

The experimental results also raise concerns. For instance, in the ablation study, even a simple decoder paired with FDA-t achieves an R^2 exceeding 40, far above the best baseline (\sim 20). This suggests the flow-matching objective is critical, but the method's core idea—learning a latent space via continuous flow and matching it between source and target domains—seems conceptually similar to other approaches like ERDiff. While the new objective may offer some advantages, the consistent outperforming of baselines by such a large margin raises the possibility of information leakage. If not, the substantial improvements require rigorous validation, including comprehensive ablation studies comparing different dynamics learning methods and the overall framework. Additionally, unlike most other methods, which employ generative models, this paper does not. Is this the key difference underlying the improvements? If so, the authors need to clarify and validate this point.

In conclusion, while the method is promising and exciting, the paper suffers from several issues: unsubstantiated and inconsistent claims, a lack of cohesive narrative, and insufficiently rigorous ablation studies to substantiate the results. These critical factors significantly reduce the likelihood of acceptance despite partial responses to reviewer concerns.

---

### Decision · Program_Chairs · 2025-01-22

Reject